# Pusa V1.0: Unlocking Temporal Control in Pretrained Video Diffusion Models via Vectorized Timestep Adaptation

**Yaofang Liu**[1,7†*]  **Yumeng Ren**[1,7]  **Aitor Artola**[1,7]  **Yuxuan Hu**[2,3]  **Xiaodong Cun**[4]
**Xiaotong Zhao**[5]  **Alan Zhao**[5]  **Raymond H. Chan**[6,7]  **Suiyun Zhang**[3]
**Rui Liu**[3†]  **Dandan Tu**[3†]  **Jean-Michel Morel**[1†]

[1]City University of Hong Kong  [2]The Chinese University of Hong Kong  [3]Huawei Research
[4]Great Bay University  [5]AI Technology Center, Tencent PCG  [6]Lingnan University
[7]Hong Kong Centre for Cerebro-Cardiovascular Health Engineering
[†]Corresponding authors

## Abstract

The rapid advancement of video diffusion models has been hindered by fundamental limitations in temporal modeling, particularly the rigid synchronization of frame evolution imposed by conventional scalar timestep variables. While task-specific adaptations and autoregressive models have sought to address these challenges, they remain constrained by computational inefficiency, catastrophic forgetting, or narrow applicability. In this work, we present **Pusa**[1] V1.0, a versatile model that leverages **vectorized timestep adaptation (VTA)** to enable fine-grained temporal control within a unified video diffusion framework. Note that VTA is a non-destructive adaptation, which means that it fully preserves the capabilities of the base model. Unlike conventional methods like Wan-I2V, which finetune a base text-to-video (T2V) model with abundant resources to do image-to-video (I2V), we achieve comparable results in a zero-shot manner after an ultra-efficient finetuning process based on VTA. Moreover, this method also unlocks many other zero-shot capabilities simultaneously, such as start-end frames and video extension —all without task-specific training. Meanwhile, it keeps the T2V capability from the base model. Mechanistic analyses also reveal that our approach preserves the foundation model's generative priors while surgically injecting temporal dynamics, avoiding the combinatorial explosion inherent to the vectorized timestep. This work establishes a scalable, efficient, and versatile paradigm for next-generation video synthesis, democratizing high-fidelity video generation for research and industry alike.

## 1  Introduction

Diffusion models Song et al. (2020); Ho et al. (2020) have transformed generative modeling, achieving remarkable results in image synthesis. Their extension to video generation Ho et al. (2022); He et al. (2022); Chen et al. (2023); Wang et al. (2023); Ma et al. (2024); OpenAI (2024); Xing et al. (2023b); Liu et al. (2024a) has been a focal point. However, these mainstream video diffusion models (VDMs) still employ a scalar timestep variable following image diffusion models, enforcing uniform temporal evolution across all frames. This approach, while being effective for text-to-video (T2V), struggles with nuanced, temporal-dependent tasks like image-to-video (I2V) and video extension Liu et al. (2024b); Wan et al. (2025); Xing et al. (2023a).

---

[*]Work partially done during an internship at Huawei Research.

[1]Pusa (/puːˈsɑː/) normally refers to "Thousand-Hand Guanyin" in Chinese, reflecting the iconography of many hands to symbolize boundless compassion and ability. We use this name to indicate that our model uses many timestep variables to achieve various video generation capabilities, and we will fully open source it to let the community benefit from this technology.

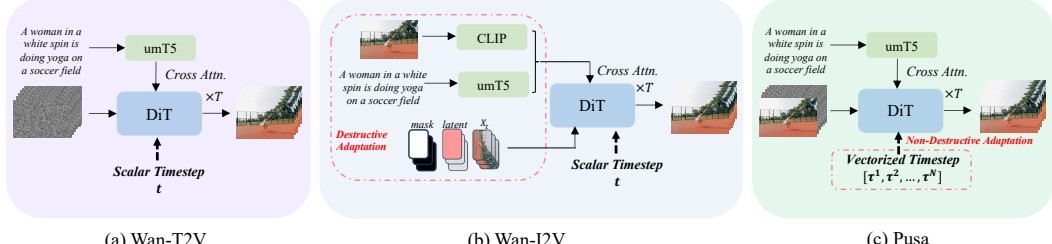

Figure 1: **Method comparison.** (b) Wan2.1-I2V-14B (Wan-I2V) and (c) Pusa both support I2V generation and are fine-tuned from (a) Wan2.1-T2V-14B (Wan-T2V). Specifically, Wan-I2V represents the mainstream practice for I2V, which modifies the base T2V model with a mask mechanism and an image clip embedding, leading to the destruction of the pretrained priors of the base model. In contrast, Pusa proposes a non-destructive VTA approach, which only inflates the model's timestep variable from a scalar to a vector. In this way, Pusa fully utilizes the pretrained priors and uses much less data and computation to achieve comparable I2V results.

Autoregressive alternatives like Diffusion Forcing Chen et al. (2024) and AR-Diffusion Sun et al. (2025) have explored avoiding this rigid synchronization modeling form of conventional VDMs. Nonetheless, their applications for video generation, like start-end frames, remained constrained by the autoregressive design. Despite large-scale deployment such as MAGI-1 Teng et al. (2025) and SkyReels V2 Chen et al. (2025) further advanced scalability, they still face challenges in computational efficiency, bidirectional reasoning, and error accumulation over long sequences.

Concurrently, FVDM Liu et al. (2024b) proposed a vectorized timestep to reframe the video diffusion paradigm fundamentally. Specifically, instead of using a scalar timestep to control the noise level of the whole video, it allows independent noise evolution per frame by assigning each frame a timestep, together results in a timestep vector for the video. Besides, to address the potential computational explosion inherent with frame-level timesteps during training, FVDM introduces a novel probabilistic timestep sampling strategy (PTSS), which not only achieves training efficiency comparable to that of conventional scalar timestep approaches but also unlocks T2V, I2V, and other temporal control tasks simultaneously.

In this work, we extend the paradigm of Frame-Aware Video Diffusion Models (FVDM)Liu et al. (2024b) to an industrial scale by proposing a vectorized timestep adaptation (VTA) strategy, which adapts pretrained large-scale VDMs to support frame-level timesteps (shown in Fig. 1). Besides, to enable fine-grained temporal control with minimal finetuning data and computation, we design the VTA to be non-destructive, which means it does not need any architectural modification to the base model and thus fully preserves the model's capabilities. As a result, our proposed model, Pusa V1.0, achieves SOTA level performance on the VBench-I2V benchmark Huang et al. (2024), comparable with Wan-I2V, which is also adapted and fine-tuned from the same base model (Wan-T2V), despite using only 4K samples and 0.5K compute cost (vs. expected $\geq$ 10M samples and $\geq$ 100K cost Wan et al. (2025)). Besides I2V capability, Pusa also generalizes to tasks such as start-end frames and video extension, all in a zero-shot way without any task-specific retraining. This highlights the great potential and versatility of the FVDM/Pusa paradigm at scale.

Our contributions can be summarized as:

- **Unprecedented Efficiency:** We present Pusa with a non-destructive VTA method to the base T2V model and an ultra-efficient finetuning strategy, which achieves SOTA I2V performance with only 4K samples and 0.5K compute cost.

- **Unified Multi-Task Generalization:** Pusa not only preserves T2V capability from the base model, but also generalizes well to many advanced temporal tasks like I2V, start-end frames, video extension, etc., all in a zero-shot way.

- **Underlying Superiority:** Mechanistic analyses reveal that our approach preserves the foundation model's generative priors while surgically injecting temporal dynamics, avoiding the combinatorial explosion inherent to vectorized timesteps. This work marks a critical shift in video generation by combining principled temporal modeling with efficient adaptation, enabling scalable, high-quality, and general-purpose video synthesis.

## 2 METHODOLOGY

### 2.1 PRELIMINARIES: FLOW MATCHING FOR GENERATIVE MODELING

Generative modeling aims to learn a model capable of synthesizing samples from a target data distribution $q_0(\mathbf{z})$ over $\mathbb{R}^D$. Continuous Normalizing Flows (CNFs) Chen (2018) achieve this by transforming samples $\mathbf{z}_1$ from a simple base distribution $q_1(\mathbf{z})$ (e.g., a standard Gaussian $\mathcal{N}(\mathbf{0}, \mathbf{I})$) to samples $\mathbf{z}_0$ that approximate the target distribution $q_0(\mathbf{z})$. This transformation is defined by an invertible mapping, often conceptualized as an ordinary differential equation (ODE) trajectory. Specifically, a probability path $\{\mathbf{z}_t\}_{t\in[0,1]}$ is defined, connecting $\mathbf{z}_0 \sim q_0$ at $t = 0$ to $\mathbf{z}_1 \sim q_1$ at $t = 1$. The dynamics along this path are described by an ODE:

$$\frac{d\mathbf{z}_t}{dt} = v_t(\mathbf{z}_t, t), \quad t \in [0, 1] \tag{1}$$

where $v_t : \mathbb{R}^D \times [0, 1] \to \mathbb{R}^D$ is a time-dependent vector field.

Flow Matching (FM) Lipman et al. (2022); Liu et al. (2022); Tong et al. (2023) is a simulation-free technique to directly learn this vector field $v_t(\mathbf{z}_t, t)$ by training a neural network $v_\theta(\mathbf{z}_t, t)$ to approximate it. This is achieved by regressing $v_\theta(\mathbf{z}_t, t)$ against a target vector field $u_t(\mathbf{z}_t|\mathbf{z}_0, \mathbf{z}_1)$. This target field is defined along specified probability paths $p_t(\mathbf{z}_t|\mathbf{z}_0, \mathbf{z}_1)$ that connect samples $\mathbf{z}_0 \sim q_0$ to corresponding samples $\mathbf{z}_1 \sim q_1$.

A common choice for these paths is a linear interpolation between a data sample $\mathbf{z}_0$ and a prior sample $\mathbf{z}_1$:

$$\mathbf{z}_t = (1 - t)\mathbf{z}_0 + t\mathbf{z}_1, \quad t \in [0, 1] \tag{2}$$

For such paths, the conditional target vector field is the time derivative of $\mathbf{z}_t$:

$$u_t(\mathbf{z}_0, \mathbf{z}_1) = \frac{d\mathbf{z}_t}{dt} = \mathbf{z}_1 - \mathbf{z}_0 \tag{3}$$

Note that for linear interpolation paths, $u_t$ is independent of $t$ and $\mathbf{z}_t$, depending only on the endpoints $\mathbf{z}_0$ and $\mathbf{z}_1$. The (conditional) flow matching objective function to train the neural network $v_\theta$ is then:

$$\mathcal{L}_{\text{FM}}(\theta) = \mathbb{E}_{t\sim\mathcal{U}[0,1],\mathbf{z}_0\sim q_0,\mathbf{z}_1\sim q_1} \left[ \|v_\theta((1 - t)\mathbf{z}_0 + t\mathbf{z}_1, t) - (\mathbf{z}_1 - \mathbf{z}_0)\|_2^2 \right] \tag{4}$$

where $\mathcal{U}[0, 1]$ is the uniform distribution over $[0, 1]$ and $\|\cdot\|_2^2$ denotes the squared Euclidean norm. Once $v_\theta$ is trained, new samples that approximate $q_0$ can be generated by first sampling $\mathbf{z}_1 \sim q_1$ and then solving the ODE $\frac{d\mathbf{z}_t}{dt} = v_\theta(\mathbf{z}_t, t)$ from $t = 1$ down to $t = 0$. The resulting $\mathbf{z}_0$ is a generated sample.

### 2.2 FRAME-AWARE FLOW MATCHING

Flow matching has become the common choice for SOTA video generation models Wan et al. (2025); Kong et al. (2024); Team (2024). To enable nuanced temporal modeling in existing SOTA models, we first need to extend the FVDM Liu et al. (2024b) paradigm to the flow matching framework.

A video clip $\mathbf{X}$ is represented as a sequence of $N$ frames. Each frame $\mathbf{x}^i \in \mathbb{R}^d$ is a $d$-dimensional column vector. The entire video clip can be represented as an $N \times d$ matrix $\mathbf{X}$, where the $i$-th row is $\mathbf{x}^{i\top}$. This can be written as $\mathbf{X} = [\mathbf{x}^1, \mathbf{x}^2, \ldots, \mathbf{x}^N]^\top$, thus $\mathbf{X} \in \mathbb{R}^{N\times d}$.

In contrast to the single-scalar time variable $t$ used in standard flow matching (Eq. 4), we introduce a **vectorized timestep variable (VTV)** $\boldsymbol{\tau} \in [0, 1]^N$, defined as:

$$\boldsymbol{\tau} = [\tau^1, \tau^2, \ldots, \tau^N]^\top \tag{5}$$

Here, each component $\tau^i \in [0, 1]$ represents the individual progression parameter of the $i$-th frame along its respective probability path from the data distribution to a prior distribution. This vectorization allows each frame to evolve at a potentially different rate or stage within the generative process.

Let $\mathbf{X}_0 = [\mathbf{x}_0^1, \ldots, \mathbf{x}_0^N]^\top$ be a video sampled from the true data distribution $q_{\text{data}}(\mathbf{X})$, where $\mathbf{x}_0^i \in \mathbb{R}^d$. Similarly, let $\mathbf{X}_1 = [\mathbf{x}_1^1, \ldots, \mathbf{x}_1^N]^\top$ be a video sampled from a simple prior distribution $q_{\text{prior}}(\mathbf{X})$ (e.g., each frame $\mathbf{x}_1^i$ is drawn independently from $\mathcal{N}(\mathbf{0}, \sigma^2\mathbf{I}_d)$).

For each frame $i$, we define a conditional probability path $p(\mathbf{x}^i_{\tau^i}|\mathbf{x}^i_0, \mathbf{x}^i_1)$ indexed by its individual timestep $\tau^i$. Adopting the linear interpolation strategy from Eq. 2 for each frame (which are $d$-dimensional vectors):

$$\mathbf{x}^i_{\tau^i} = (1 - \tau^i)\mathbf{x}^i_0 + \tau^i \mathbf{x}^i_1 \tag{6}$$

The state of the entire video, corresponding to a specific vectorized timestep $\boldsymbol{\tau}$, is then given by the $N \times d$ matrix $\mathbf{X}_{\boldsymbol{\tau}}$, whose $i$-th row is $\mathbf{x}^{i\top}_{\tau^i}$:

$$\mathbf{X}_{\boldsymbol{\tau}} = [\mathbf{x}^1_{\tau^1}, \mathbf{x}^2_{\tau^2}, \dots, \mathbf{x}^N_{\tau^N}]^\top \tag{7}$$

We aim to learn a single neural network $v_\theta(\mathbf{X}, \boldsymbol{\tau})$ that models the joint dynamics of all frames conditioned on their respective timesteps. This network takes the current video state $\mathbf{X} \in \mathbb{R}^{N \times d}$ (which is $\mathbf{X}_{\boldsymbol{\tau}}$ during training) and the vectorized timestep $\boldsymbol{\tau} \in [0,1]^N$ as input. It outputs a velocity field for the entire video, an $N \times d$ matrix denoted as $v_\theta(\mathbf{X}, \boldsymbol{\tau}) = [\mathbf{v}^1, \dots, \mathbf{v}^N]^\top$, where each $\mathbf{v}^i \in \mathbb{R}^d$ is the velocity vector for the $i$-th frame. Thus, $v_\theta : \mathbb{R}^{N \times d} \times [0,1]^N \to \mathbb{R}^{N \times d}$.

The target vector field for the entire video $\mathbf{X}_{\boldsymbol{\tau}}$, conditioned on the initial video $\mathbf{X}_0$ and target prior $\mathbf{X}_1$, is an $N \times d$ matrix $\mathbf{U}(\mathbf{X}_0, \mathbf{X}_1)$. Its $i$-th row is the transpose of the derivative of the $i$-th frame's path (Eq. 6) with respect to its individual timestep $\tau^i$. Using the derivative from Eq. 3 for each frame:

$$\frac{d\mathbf{x}^i_{\tau^i}}{d\tau^i} = \mathbf{x}^i_1 - \mathbf{x}^i_0 \tag{8}$$

Thus, the target video-level vector field is:

$$\mathbf{U}(\mathbf{X}_0, \mathbf{X}_1) = [(\mathbf{x}^1_1 - \mathbf{x}^1_0), \dots, (\mathbf{x}^N_1 - \mathbf{x}^N_0)]^\top = \mathbf{X}_1 - \mathbf{X}_0 \tag{9}$$

Notably, for the linear interpolation path, this target vector field $\mathbf{X}_1 - \mathbf{X}_0$ is independent of both the current video state $\mathbf{X}_{\boldsymbol{\tau}}$ and the vectorized timestep $\boldsymbol{\tau}$ itself, simplifying the regression target. The video state $\mathbf{X}_{\boldsymbol{\tau}}$ at timestep $\boldsymbol{\tau}$ is constructed via frame-wise linear interpolation:

$$\mathbf{X}_{\boldsymbol{\tau}} = (1 - \boldsymbol{\tau}) \odot \mathbf{X}_0 + \boldsymbol{\tau} \odot \mathbf{X}_1 \tag{10}$$

where $\odot$ denotes element-wise multiplication between the timestep vector $\boldsymbol{\tau} = [\tau^1, \tau^2, ..., \tau^N]^\top$ and each frame.

**Key Properties:**

1. *Path Consistency*: Each frame evolves linearly: $\mathbf{x}^i_{\tau^i} = (1 - \tau^i)\mathbf{x}^i_0 + \tau^i \mathbf{x}^i_1$

2. *Vector Field Simplicity*: $\frac{d\mathbf{X}_{\boldsymbol{\tau}}}{d\boldsymbol{\tau}} = \mathbf{X}_1 - \mathbf{X}_0$ (constant for all $\boldsymbol{\tau}$)

3. *Decoupling*: Frame dynamics depend only on their own $\tau^i$, enabling asynchronous evolution.

The parameters $\theta$ of the neural network $v_\theta$ are optimized by minimizing the Frame-Aware Flow Matching (FAFM) objective function:

$$\mathcal{L}_{\text{FAFM}}(\theta) = \mathbb{E}_{\mathbf{X}_0 \sim q_{\text{data}}, \mathbf{X}_1 \sim q_{\text{prior}}, \boldsymbol{\tau} \sim p_{\text{PTSS}}(\boldsymbol{\tau})} \left[ \|v_\theta(\mathbf{X}_{\boldsymbol{\tau}}, \boldsymbol{\tau}) - (\mathbf{X}_1 - \mathbf{X}_0)\|^2_F \right] \tag{11}$$

where $\mathbf{X}_{\boldsymbol{\tau}}$ is the video state constructed according to Eq. 7, $\|\cdot\|^2_F$ denotes the squared Frobenius norm, $\boldsymbol{\tau} \sim p_{\text{PTSS}}(\boldsymbol{\tau})$ indicates that the vectorized timestep $\boldsymbol{\tau}$ is sampled according to PTSS in FVDM Liu et al. (2024b). This strategy is designed to expose the model to both asynchronous frame evolutions with a probability $p_{\text{async}} \in [0,1]$ and synchronized frame evolutions with probability $1 - p_{\text{async}}$ during training.

## 2.3 VECTORIZED TIMESTEP ADAPTATION

In this work, we aim to adapt a large-scale, pre-trained T2V diffusion models to support the vectorized timestep Liu et al. (2024b). The adaptation, which we term Vectorized Timestep Adaptation (VTA), along with a lightweight fine-tuning process, imbues the model with fine-grained temporal control, enabling advanced capabilities such as zero-shot I2V.

### 2.3.1 Implementation of Vectorized Timestep Adaptation

The foundational principle of our implementation is to re-engineer the core architecture to process a vectorized timestep $\boldsymbol{\tau}$ instead of a scalar timestep $t$. The architectural adaptation is primarily focused on the model's temporal conditioning mechanism. We introduce two key modifications:

**Vectorized Timestep Embedding:** The timestep embedding module is modified to process the input vector $\boldsymbol{\tau}$, generating a sequence of frame-specific embeddings $\mathbf{E}_{\boldsymbol{\tau}} \in \mathbb{R}^{N_1 \times D}$, where $N_1$ is the number of frames in the video latent sequence, each vector in the sequence corresponds to a latent frame's individual.

**Per-Frame Modulation:** These frame-specific embeddings are subsequently projected to produce per-frame modulation parameters (i.e., scale, shift, and gate) within each block of the DiT architecture. The operation of a DiT Peebles & Xie (2023) block on the latent representation $\mathbf{z}^i$ of the $i$-th frame is thus conditioned on its individual timestep $\tau^i$, which can be conceptually expressed as:

$$\mathbf{z}_{\text{out}}^i = \text{DiTBlock}(\mathbf{z}_{\text{in}}^i, \text{context}, \text{modulate}(\tau^i))$$

Note that this modification is non-destructive, which means it fully preserves the T2V capability of the base model. The adapted model generates the same results by setting all frame timesteps to the same values as the base model.

Table 1: **Vbench-I2V results.** Our model demonstrates SOTA-level performance, achieving a top-tier rank among open-source models, and is notably comparable with its architectural baseline, Wan-I2V. All scores are reported in percentages (%). Higher is better for all metrics. Best score in each column is in **bold**. **Abbreviations**: **SC**: Subject Consistency, **BC**: Background Consistency, **MS**: Motion Smoothness, **DD**: Dynamic Degree, **AQ**: Aesthetic Quality, **IQ**: Imaging Quality, **I2V-S**: I2V Subject Consistency, **I2V-B**: I2V Background Consistency, **CM**: Camera Motion.

| Model | Overall Scores ↑ | | | Quality Metrics ↑ | | | | | | I2V Metrics ↑ | | |
|---|---|---|---|---|---|---|---|---|---|---|---|---|
| | Total | I2V | Quality | SC | BC | MS | DD | AQ | IQ | I2V-S | I2V-B | CM |
| *— Closed / Proprietary Models —* | | | | | | | | | | | | |
| Gen-4-I2V (API) | 88.27 | 95.65 | 80.89 | 93.23 | 96.79 | 98.99 | 55.20 | 61.77 | 70.41 | 97.84 | 97.46 | **68.26** |
| STIV (Apple) | 86.73 | 93.48 | 79.98 | 98.40 | 98.39 | **99.61** | 15.28 | 66.00 | 70.81 | **98.96** | 97.35 | 11.17 |
| *— Open Source Models —* | | | | | | | | | | | | |
| Magi-1 | **89.28** | **96.12** | **82.44** | 93.96 | 96.74 | 98.68 | 68.21 | 64.74 | 69.71 | 98.39 | **99.00** | 50.85 |
| Step-Video-TI2V | 88.36 | 95.50 | 81.22 | 96.02 | 97.06 | 99.24 | 48.78 | 62.29 | 70.44 | 97.86 | 98.63 | 49.23 |
| DynamiCrafter-512 | 86.99 | 93.53 | 80.46 | 93.81 | 96.64 | 96.84 | **69.67** | 60.88 | 68.60 | 97.21 | 97.40 | 31.98 |
| CogVideoX-5b-I2V | 86.70 | 94.79 | 78.61 | 94.34 | 96.42 | 98.40 | 33.17 | 61.87 | 70.01 | 97.19 | 96.74 | 67.68 |
| Animate-Anything | 86.48 | 94.25 | 78.71 | **98.90** | 98.19 | 98.61 | 2.68 | **67.12** | **72.09** | 98.76 | 98.58 | 13.08 |
| SEINE-512x512 | 85.52 | 92.67 | 78.37 | 95.28 | 97.12 | 97.12 | 27.07 | 64.55 | 71.39 | 97.15 | 96.94 | 20.97 |
| I2VGen-XL | 85.28 | 92.11 | 78.44 | 94.18 | 97.09 | 98.34 | 26.10 | 64.82 | 69.14 | 96.48 | 96.83 | 18.48 |
| ConsistI2V | 84.07 | 91.91 | 76.22 | 95.27 | 98.28 | 97.38 | 18.62 | 59.00 | 66.92 | 95.82 | 95.95 | 33.92 |
| VideoCrafter | 82.57 | 86.31 | 78.84 | 97.86 | **98.79** | 98.00 | 22.60 | 60.78 | 71.68 | 91.17 | 91.31 | 33.60 |
| CogVideoX1.5-5B | 71.58 | 92.25 | 50.90 | 91.80 | 94.66 | 40.98 | 62.29 | 70.21 | 67.07 | 96.46 | 95.50 | 39.71 |
| SVD-XT-1.1 | – | – | 79.40 | 95.42 | 96.77 | 98.12 | 43.17 | 60.23 | 70.23 | 97.51 | 97.62 | – |
| SVD-XT-1.0 | – | – | 80.11 | 95.52 | 96.61 | 98.09 | 52.36 | 60.15 | 69.80 | 97.52 | 97.63 | – |
| **Wan-I2V** | 86.86 | 92.90 | **80.82** | **94.86** | 97.07 | 97.90 | 51.38 | **64.75** | 70.44 | 96.95 | 96.44 | **34.76** |
| **Ours** | **87.32** | **94.84** | 79.80 | 92.27 | 96.02 | **98.49** | 52.60 | 63.15 | 68.27 | **97.64** | **99.24** | 29.46 |

### 2.3.2 Training Procedure

The optimization follows the FAFM objective defined in Eq. 11. A key advantage of our approach is its simplicity: **by leveraging the robust generative prior of the base model, we circumvent the need for sampling synchronous timesteps. Instead, we train the model directly with a fully randomized vectorized timestep** ($p_{\text{async}} = 1$), where each component $\tau^i$ is sampled independently from $U[0, 1]$. This stochastic training regimen compels the model to learn fine-grained temporal control from a maximally diverse distribution of temporal states.

### 2.3.3 Inference for Image-to-Video

Pusa performs zero-shot I2V generation by strategically manipulating the vectorized timestep $\boldsymbol{\tau}$ during sampling. To condition generation on a starting image $I_0$, for simplicity and fair comparison

with baselines, we clamp its timestep component to zero throughout inference (i.e., $\tau_s^1 = 0$ for all steps $s$). Note that we can also add some noise (e.g., set $\tau_s^1 = 0.2 * s$ or any level of noise) to the first frame, which may synthesize more coherent videos with a slight change to the first frame. During sampling, which follows the Euler method for ODE integration, this ensures the change in the first frame's latent is always zero, effectively fixing it as a clean condition. This flexible control scheme naturally extends to other complex temporal tasks, such as start-end frames and video extension. The detailed I2V sampling algorithm is outlined in Appendix B.

## 3 EXPERIMENTS

Our experiments are designed to rigorously validate the three core contributions of this work: (1) the unprecedented efficiency and SOTA-level performance of our model, Pusa, on the primary task of I2V generation; (2) the underlying mechanism for why Pusa works; and (3) the emergent zero-shot multi-task capabilities of Pusa.

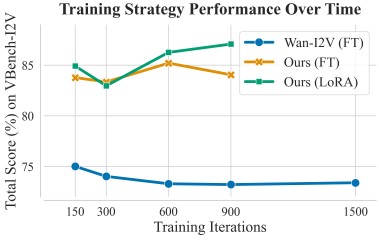

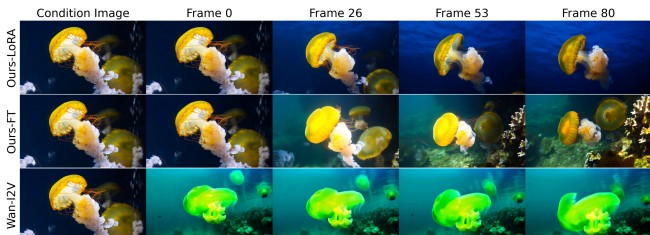

Figure 2: **Quantitative comparison of training strategies.** Both our LoRA and full fine-tuning (FT) approaches consistently outperform Wan-I2V's fine-tuning method, achieving much superior scores with unprecedented efficiency.

Figure 3: **Qualitative comparison of training strategies at 900 iterations.** From top to bottom: Ours (LoRA, $\alpha = 1.4$), Ours (FT), and Wan-I2V like baseline. Our methods maintain high fidelity to the condition image, whereas the baseline fails to preserve subject identity and color under the training budget.

### 3.1 SETUP

Our method works for both full fine-tuning and Lora fine-tuning. Towards adaptation with fewer GPUs, we perform fine-tuning on the SOTA open-source Wan-T2V model using the LoRA (Low-Rank Adaptation) technique Hu et al. (2022) for our final model, which enables parameter-efficient training. The training infrastructure consists of 8 GPUs, each has 80GB of memory and high bandwidth, with DeepSpeed Zero2 Rajbhandari et al. (2020) for memory optimization, achieving a total batch size of 8. The LoRA implementation is based on DiffSynth-Studio[2], leveraging its optimized diffusion model training pipeline. Regarding the fine-tuning dataset, we directly utilize T2V samples generated by Wan-T2V Zheng et al. (2025); Wan et al. (2025). This dataset includes 3,860 high-quality 720p videos and spans diverse visual domains and temporal structures (e.g., natural scenes, human activities, camera motion), ensuring robust model generalization and aligned with Wan's original distribution. For evaluation, we use Vbench-I2V Huang et al. (2024) for comprehensive I2V capability evaluation. Note that there is no overlap between the training and evaluation data. For baseline comparison, we use the whole testing set, generating 5590 videos. For hyperparameter and ablation studies, we test with a subset of 750 videos.

### 3.2 BASELINE COMPARISON

As shown in Table 1, Pusa, with only 10 inference steps, achieves SOTA-level performance among open-source models. This result is also comparable to the result of its direct baseline, Wan-I2V. Note that all the baselines are trained with vastly greater resources. More specifically, our model

---

[2]https://github.com/modelscope/DiffSynth-Studio

obtains a total score of 87.32, outperforming Wan-I2V's 86.86. Besides, Pusa demonstrates superior performance in key I2V metrics, such as I2V Background Consistency (99.24 vs. 96.44) and I2V Subject Consistency (97.64 vs. 96.95), indicating a more faithful adherence to the input image condition. Furthermore, our model exhibits a higher Dynamic Degree (52.60 vs. 51.38), producing more motion-rich videos while maintaining high Motion Smoothness (98.49 vs. 97.90). Hyperparameter studies about Pusa are available in Appendix C.1.

## 3.3 ABLATION STUDY

We conduct a series of ablation studies to dissect the core components and verify the effectiveness of our proposed methodology. Due to space constraints, we present a summary of our key findings here. Comprehensive results are provided in the Appendix C.2.

**Training Strategy.** We compare three distinct training strategies: the baseline model trained with the Wan-I2V method, our approach with full fine-tuning, and LoRA fine-tuning on the same dataset with 480p resolution. As illustrated in Fig. 2, both our methods converge at a very early stage with close results and substantially outperform the baseline across all training steps. This quantitative superiority is mirrored in our qualitative results (Fig. 3). At 900 iterations, both our LoRA and fully fine-tuned models generate videos highly faithful to the conditioning image. In stark contrast, the baseline fails catastrophically under the same training budget, with the generated video completely diverging from the source. This underscores a critical finding: the Wan-I2V's method is profoundly inefficient, requiring a massive scale of data and compute to reach its official performance Wan et al. (2025). Our method, conversely, delivers superior results with a minuscule fraction of these resources, demonstrating a paradigm shift in training efficiency.

**Timestep Sampling Strategy.** Our framework employs a simple yet powerful strategy of sampling purely random timesteps for each frame during training. To validate this design, we compare it against a more structured approach, PTSS, where timesteps are preferentially sampled from a specific range, and an intuitive baseline for I2V with the first frame always being noise-free $\tau^1 = 0$ while all other timesteps being of the same value. Table 2 summarizes the performance at 900 steps. The results unequivocally demonstrate that our fully random sampling strategy achieves the highest scores, surpassing all baselines. This finding confirms that our training strategy is more effective for learning robust temporal dynamics in our framework.

Table 2: **Ablation on Timestep Sampling Strategy.** Comparison of overall scores at 900 training iterations. Our random sampling approach achieves the highest performance.

| Sampling Strategy | Total (%) | Quality (%) | I2V (%) |
|---|---|---|---|
| Ours | **87.69** | **80.55** | **94.83** |
| I2V | 73.27 | 69.96 | 76.57 |
| PTSS ($p = 0.2$) | 84.74 | 77.60 | 91.88 |
| PTSS ($p = 0.8$) | 86.49 | 79.30 | 93.69 |

Table 3: **Ablation Study on Base Model.**

| Base Model | Setting | Overall | | | Quality Metrics | | | | | | I2V Metrics | | |
|---|---|---|---|---|---|---|---|---|---|---|---|---|---|
| | | Total | Quality | I2V | SC | BC | MS | DD | AQ | IQ | I2V-S | I2V-B | CM |
| **Wan2.1** | $\alpha = 1.4$ | **87.69** | **80.55** | 94.83 | 91.39 | 96.02 | **98.10** | **66.40** | **62.42** | 68.57 | **97.78** | **99.33** | 26.40 |
| **Wan2.2** | high $\alpha = 1.5$, low $\alpha = 1.4$ | **87.69** | 79.89 | **95.49** | **91.57** | **96.83** | 97.91 | 63.60 | 59.30 | **68.75** | 96.99 | 99.18 | **51.60** |

**Base Model Comparison.** We further investigate the impact of different base models on final performance. As shown in Table 3, both our Wan2.1-based model and our Wan2.2-based model achieve an identical overall score of 87.69. However, they demonstrate different characteristic strengths: the Wan2.1 variant excels in aesthetic quality and dynamic degree. Conversely, the Wan2.2 variant generates videos with significantly more pronounced camera motion (51.60 vs. 26.40). Since we use the same dataset for finetuning, these differences mainly come from the base models themselves. Overall, this suggests our method works well with different base models, and the choice of base model can be tailored to application needs and the model's characteristics.

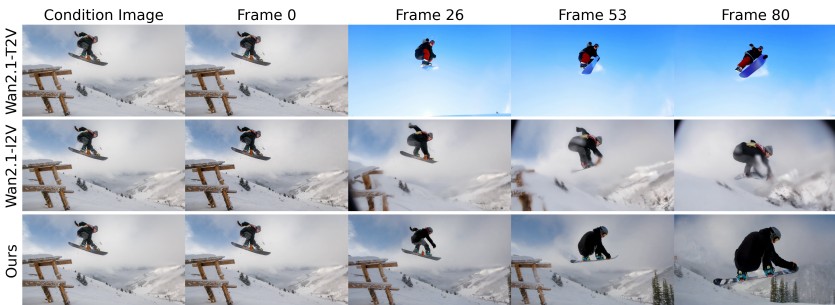

Figure 4: I2V results, where our model generates a smooth and realistic animation of the first frame while Wan2.1-T2V generates completely different frames but aligned with the text prompt, and Wan2.1-I2V generates frames with noticeable distortions. All videos are generated with the same condition image and text prompt: "a snowboarder is in the air doing a trick".

## 3.4 ANALYSIS OF THE ADAPTATION MECHANISM

We investigate the mechanisms underlying Pusa's efficient adaptation, showing that it performs highly targeted parameter updates that augment — rather than overwrite — the pretrained knowledge of the foundation model.

**I2V Qualitative.** In Fig. 4, Wan-T2V with VTA uses the same inference method as Pusa to do I2V generation, which is by directly setting the first frame noise-free. Since the model is only trained for the T2V task, the generated frames only align with the text prompt but have no relation to the first frame. Meanwhile, Pusa achieves seamless alignment with both text and input image, outperforming Wan-I2V, which exhibits visible distortions and poor character preservation.

**Attention Mechanism.** Visualization of frame-to-frame self-attention maps (Fig. 5) reveals critical differences. Specifically, we plot the self-attention maps of queries and keys within the final Transformer block across 3 different inference steps, i.e., steps 0, 4, and 9 (10 steps in total). Wan-T2V exhibits a diagonal attention pattern, indicating that each frame primarily attends to itself, with little frame correlation. In contrast, both Wan-I2V and Pusa show strong attention from all frames to the first frame initially, which is essential for maintaining consistency with the input image. However, a key difference emerges: the attention to the first frame in Wan-I2V is only strengthened in step 0. In Pusa, the attention to the first frame is significantly enhanced across all steps. This exemplifies Pusa's surgical injection of temporal dynamics.

**Parameter Divergence.** This observation is further supported by an analysis of parameter changes (Fig. 6). The parameter drift in Wan-I2V is substantial and concentrated in modules critical for content generation, such as the text encoder and cross-attention blocks. This implies a significant alteration of the model's core generative priors. Pusa, in contrast, exhibits minimal parameter changes, with modifications almost exclusively in the self-attention blocks responsible for temporal dynamics. The magnitude of parameter change in Wan-I2V is more than an order of magnitude larger than in Pusa. This confirms that our approach constitutes a minimal, targeted adaptation, preserving the integrity of the foundation model, and explaining its efficiency.

**Vectorized Timestep Adaptation Efficacy.** The VTV framework faces a combinatorial explosion in temporal composition space (e.g., $10^{48}$ configurations for 16 frames), making convergence from scratch challenging. Pusa circumvents this by leveraging Wan-T2V's pretrained video generation capabilities, requiring only brief fine-tuning to master temporal control with independent timesteps. The base model's inherent robustness to timestep asynchronization is evidenced by its coherent zero-shot I2V generation (Fig. 4), despite failing image-condition adherence. This stems from Wan-T2V's diagonal self-attention patterns (Fig. 5), indicating frame synthesis independence. Pusa's fine-tuning surgically introduces targeted temporal correlation while preserving the stable generative core, enabling nondestructive adaptation that solves the VTV compositionality problem with unprecedented efficiency.

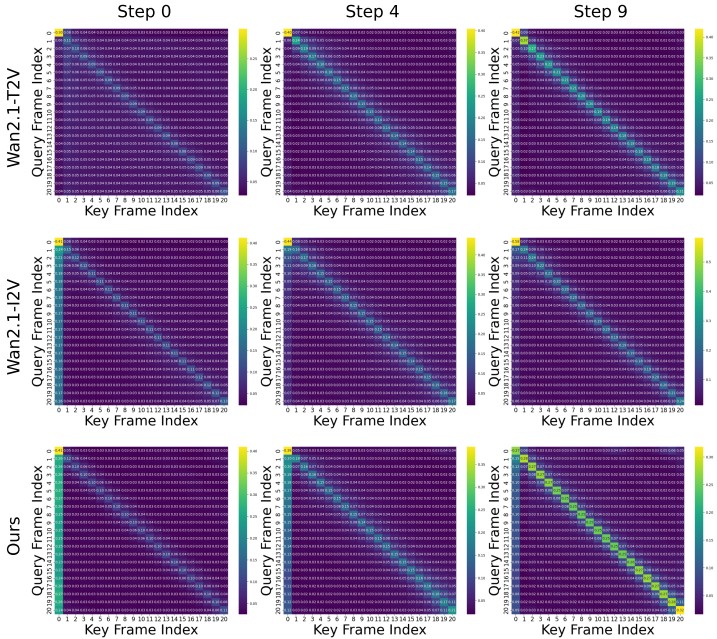

Figure 5: Visualization of attention maps. Specifically, these maps are of the last DiT block across multiple inference steps. Each value in the attention map represents frame-to-frame correlation/attention; a larger value means higher correlation. Zoom in for a better view.

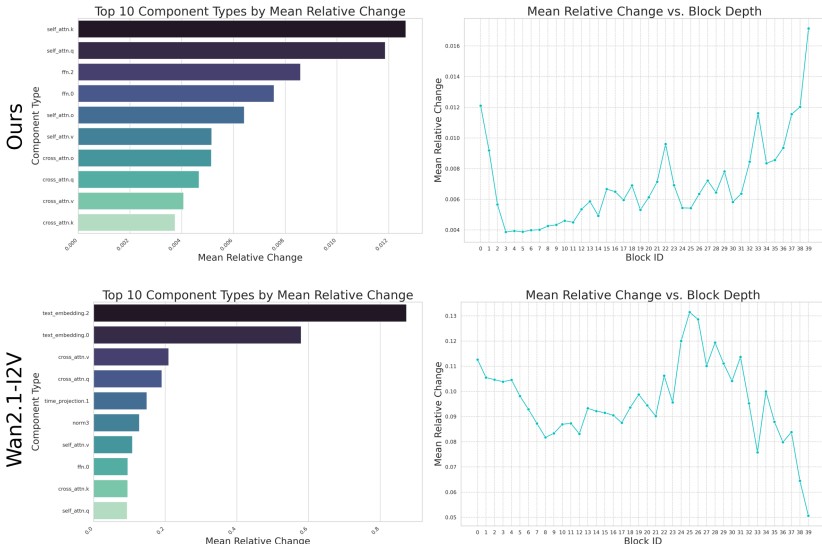

Figure 6: Analysis on finetuned model's parameter shifts. The columns, from left to right, represent the top 10 components with largest average relative parameter changes and average change by transformer blocks, respectively. Zoom in for a better view.

## 3.5 ZERO-SHOT MULTI-TASK CAPABILITIES

Our approach demonstrates exceptional generalization across diverse video generation tasks beyond T2V without task-specific training. This capability comes from flexible vectorized time step settings, enabling arbitrary conditioning on any subset of frames with any level of noise. Besides, Pusa retains high T2V generation quality after adaptation from its strong foundation model, confirming that our fine-tuning process avoids catastrophic forgetting of the primary task. More significantly, the proposed method exhibits remarkable zero-shot performance on complex temporal synthesis tasks, including I2V generation, start-end frames, video completion, video extension, and so on. Comprehensive results for these tasks are available in Appendix D.

## 4 CONCLUSION

This work introduces Pusa, which achieves SOTA-level I2V performance based on Wan-T2V with unprecedented efficiency, requiring only $500 and 4K samples. The key innovation lies in our non-destructive VTA strategy, preserving the foundation model's robust priors while enabling frame-independent evolution. This unlocks many zero-shot generalizations to diverse tasks, a property unmatched by conventional or auto-regressive VDMs. Our analysis demonstrates that Pusa's success stems from its minimal, targeted modifications to the base model's temporal attention mechanisms, avoiding the catastrophic forgetting observed in task-specific fine-tuning. The implications are profound: Pusa redefines the efficiency-quality tradeoff in video generation, enabling high-fidelity, multi-task generation at a fraction of traditional costs.

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

APPENDIX

# A    MORE RELATED WORKS

The field of video generation has rapidly evolved, driven by the success of diffusion models in image synthesis. Our work builds upon and extends several key research threads, positioning itself as a highly efficient and versatile solution for the next paradigm of video diffusion models.

## A.1    CONVENTIONAL VIDEO DIFFUSION MODELS

The initial extension of image diffusion models to video generation established a foundational paradigm. Seminal works like VDM Ho et al. (2022) and subsequent large-scale models such as Latent Video Diffusion Models (LVDM) He et al. (2022), VideoCrafter1 Chen et al. (2023), and others Wang et al. (2023); Ma et al. (2024); Wan et al. (2025); Kong et al. (2024) all adopted the same diffusion framework. A core characteristic of these conventional models is their reliance on a scalar timestep variable. This single variable governs the noise level and evolution trajectory uniformly across all frames of a video clip during the diffusion process. While this synchronized-frame approach proved effective for generating short, self-contained clips, particularly for T2V tasks, it imposes a rigid temporal structure. The uniform noise schedule inherently limits the model's ability to handle tasks requiring asynchronous frame evolution, such as I2V generation, where the first frame or condition image is given, or complex editing tasks like video interpolation. Recognizing the limitations of conventional VDMs in temporal modeling, the research community has developed numerous extensions targeting specific video generation tasks Xing et al. (2023b). These approaches predominantly focus on adapting existing scalar timestep based models through fine-tuning strategies or zero-shot techniques to handle domain-specific challenges such as image-to-video generation Xing et al. (2023a); Guo et al. (2023); Zhang et al. (2023); Li et al. (2024); Ni et al. (2024), video interpolation Wang et al. (2024b;a); Wan et al. (2025), and long video synthesis Duan et al. (2024); Henschel et al. (2024); Kim et al. (2024); Lu et al. (2024); Dalal et al. (2025); Zhao et al. (2025) These methods typically involve extensive fine-tuning of a large, pre-trained T2V model on task-specific data or employing zero-shot domain transfer techniques.

I2V generation has emerged as a particularly active area. For example, the Wan-I2V model presented in the Wan paper required fine-tuning Wan T2V model on its T2V pretraining dataset to achieve its SOTA I2V capabilities and can only do I2V after this process Wan et al. (2025). Overall, these extensions reveal fundamental challenges in balancing flexibility, generalization, and the retention of original model capabilities. Fine-tuning approaches often suffer from catastrophic forgetting, where adaptation to specific tasks severely degrades performance on its original capabilityPan et al. (2024); Ramasesh et al. (2021). Zero-shot methods, exemplified by TI2V-Zero introduces a zero-shot method for conditioning T2V diffusion models on images Ni et al. (2024). However, its generalization and generation quality are limited by potential visual artifacts and reduced robustness, as its simple "repeat-and-slide" strategy struggles with diverse input and can produce blurry or flickering videos. The reliance on task-specific architectures and training procedures highlights the need for a more unified and general approach that can handle diverse video generation scenarios without requiring extensive finetuning. Our work departs from these limitations by adopting the VTV introduced by FVDM Liu et al. (2024b), enabling fine-grained control over the generative process.

## A.2    AUTOREGRESSIVE VIDEO DIFFUSION MODELS

Recently, people have explored autoregressive paradigms for video diffusion models, where frames are generated sequentially rather than simultaneously (Chen et al., 2024; Sun et al., 2025; Teng et al., 2025; Chen et al., 2025; Huang et al., 2025). This direction includes methods like Diffusion Forcing, which trains a causal next-token model to predict one or multiple future tokens without fully diffusing past ones, enabling variable-length generation capabilities. CauseVid Yin et al. (2024) represents another significant advancement in this direction, proposing fast autoregressive video diffusion models that can generate frames on-the-fly with streaming capabilities.

Large-scale autoregressive models such as MAGI-1 Teng et al. (2025) and SkyReels-V2 Chen et al. (2025) have demonstrated the potential for scalable video generation through chunk-by-chunk pro-

cessing, where each segment is denoised holistically before proceeding to the next. Self-Forcing Huang et al. (2025) addresses the critical issue of exposure bias in autoregressive video diffusion by introducing a training paradigm where models condition on their own previously generated outputs rather than ground-truth frames. This approach enables real-time streaming video generation while maintaining temporal coherence through innovative key-value caching mechanisms.

Despite these advances, autoregressive video diffusion models face inherent limitations that constrain their applicability. The sequential nature of generation restricts these models to unidirectional tasks, making them inadequate for many scenarios, such as start-end frames, video transitions, and keyframe interpolation. Moreover, error accumulation and drift issues represent persistent challenges in autoregressive approaches, where small prediction errors compound over time, leading to quality degradation in longer sequences Huang et al. (2025). Recent theoretical analyses have also identified both error accumulation and memory bottlenecks as fundamental phenomena in autoregressive video diffusion models, revealing a Pareto frontier between these competing constraints Wang et al. (2025).

### A.3 FRAME-AWARE VIDEO DIFFUSION MODEL

Frame-aware video diffusion model (FVDM) Liu et al. (2024b) is a parallel line of research to reconstruct the paradigm for video diffusion models, by enabling independent temporal evolution for each frame. The vectorized timestep approach enables unprecedented flexibility across multiple video generation tasks, including T2V, I2V, start-end frames, video extension, and so on, all within a single unified framework. Unlike conventional approaches that require extensive destructive architectural modifications and retraining, FVDM demonstrates strong zero-shot capabilities across diverse temporal conditioning scenarios with its PTSS training strategy. In this work, Pusa-Wan (V1.0) further extends FVDM to an industrial scale and proposes a dedicated VTA strategy and post-training method, achieving remarkable efficiency gains with finetuning costs reduced to mere $500 from above $100K$ of Wan-I2V, while outperforming it on Vbench-I2V.

The FVDM/Pusa framework represents a fundamental departure from previous temporal modeling approaches, offering a solution that can perform both directional generation tasks (like autoregressive models) and bidirectional temporal reasoning tasks that autoregressive approaches cannot handle. This unified capability, combined with the demonstrated computational efficiency and strong empirical results, positions this approach as a promising direction for next-generation video diffusion models.

---

**Algorithm 1** Pusa: Sampling for I2V Generation

---

**Require:** Trained model $v_\theta$, VAE Encoder $E$ and Decoder $D$, Scheduler.
**Require:** Initial image $I_0$, text prompt $c$, number of frames $N_1$, inference steps $S$.
1: Encode prompt: $\mathbf{c}_{emb} \leftarrow \text{EncodePrompt}(c)$.
2: Encode image to initial latent: $\hat{\mathbf{z}}_1^1 \leftarrow E(I_0)$.
3: Sample noise for remaining frames: $[\mathbf{z}_1^1, \mathbf{z}_1^2, \ldots, \mathbf{z}_1^{N_1}] \sim \mathcal{N}(\mathbf{0}, \mathbf{I})$.
4:          ▷ Initialize video with clean first frame and noisy subsequent frames.
5: Construct initial latent video: $\mathbf{Z}_1 \leftarrow [\hat{\mathbf{z}}_1^1, \mathbf{z}_1^2, \ldots, \mathbf{z}_1^{N_1}]$.
6: Retrieve scheduler noise levels $\{\sigma_s\}_{s=1}^S$, where $\sigma_1 > \cdots > \sigma_S \approx 0$.
7: **for** $s \leftarrow 1, \ldots, S-1$ **do**
8:      Let $\sigma_{current} \leftarrow \sigma_s$ and $\sigma_{next} \leftarrow \sigma_{s+1}$.
9:          ▷ Construct vectorized timestep to freeze the first frame.
10:      Set current path parameter $\boldsymbol{\tau}_s \leftarrow \sigma^{-1}([0, \sigma_{current}, \ldots, \sigma_{current}]^\top)$.
11:      Set current noise levels $\boldsymbol{\sigma}_s \leftarrow [0, \sigma_{current}, \ldots, \sigma_{current}]^\top$.
12:      Set next noise levels $\boldsymbol{\sigma}_{s+1} \leftarrow [0, \sigma_{next}, \ldots, \sigma_{next}]^\top$.
13:      Predict vector field: $\hat{\mathbf{U}}_s \leftarrow v_\theta(\mathbf{Z}_s, \boldsymbol{\tau}_s, \mathbf{c}_{emb})$.
14:          ▷ Update latents; first frame remains unchanged.
15:      $\mathbf{Z}_{s+1} \leftarrow \mathbf{Z}_s + \hat{\mathbf{U}}_s \odot (\boldsymbol{\sigma}_{s+1} - \boldsymbol{\sigma}_s)$.
16: **end for**
17: Decode final latent video: $\mathbf{X}_{out} \leftarrow D(\mathbf{Z}_S)$.
18: **return** Output video $\mathbf{X}_{out}$.

---

## B  INFERENCE ALGORITHM FOR I2V GENERATION

Pusa performs zero-shot I2V generation by strategically manipulating the vectorized timestep $\tau$ during sampling. To condition generation on a starting image $I_0$, for simplicity and fair comparison with baselines, we clamp its timestep component to zero throughout inference (i.e., $\tau_s^1 = 0$ for all steps $s$). Note that we can also add some noise (e.g., set $\tau_s^1 = 0.2 * s$ or any level of noise) to the first frame, which may synthesize more coherent videos with a slight change to the first frame. During sampling, which follows the Euler method for ODE integration, this ensures the change in the first frame's latent is always zero, effectively fixing it as a clean condition. This flexible control scheme naturally extends to other complex temporal tasks, such as start-end frames and video extension. The detailed I2V sampling procedure is outlined in Algorithm 1.

Table 4: **Comprehensive studies on key hyperparameters.** This table presents a detailed analysis of our model's performance by varying training iterations, LoRA configurations, and the number of inference steps. All scores are reported in percentages (%), with higher values indicating better performance. For each ablation group, the best score per metric is highlighted in **bold**.

| Group | Setting | Overall | | | Quality Metrics | | | | | | I2V Metrics | | |
|---|---|---|---|---|---|---|---|---|---|---|---|---|---|
| | | Total | Quality | I2V | SC | BC | MS | DD | AQ | IQ | I2V-S | I2V-B | CM |
| **(a) LoRA Configurations** | | | | | | | | | | | | | |
| 256 | $\alpha = 1.0$ | 79.75 | 73.25 | 86.25 | 84.35 | 88.20 | 98.91 | 28.92 | 60.90 | 64.73 | 91.69 | 91.18 | **28.00** |
| | $\alpha = 1.4$ | 83.12 | 76.07 | 90.17 | 85.65 | 96.67 | 98.99 | **30.06** | 60.13 | 67.15 | 90.33 | **98.78** | 23.48 |
| | $\alpha = 1.7$ | **85.86** | **77.96** | **93.76** | **97.18** | 96.68 | **99.06** | 10.40 | **63.04** | **70.70** | **98.60** | 98.30 | 8.40 |
| | $\alpha = 2.0$ | 84.22 | 76.30 | 92.14 | 96.71 | 94.63 | 99.05 | 6.40 | 60.34 | 69.63 | 97.96 | 95.77 | 16.00 |
| 512 | $\alpha = 1.0$ | 80.17 | 74.35 | 85.99 | 83.16 | 86.99 | **98.05** | 62.40 | 58.44 | 62.49 | 91.23 | 90.70 | 34.40 |
| | $\alpha = 1.4$ | 86.46 | 79.79 | 93.13 | 88.64 | 93.54 | 97.68 | **78.80** | 61.12 | 67.47 | 95.72 | 97.60 | **38.40** |
| | $\alpha = 1.7$ | **87.11** | **80.42** | 93.80 | 92.93 | **95.58** | 97.72 | 62.80 | **61.78** | 70.34 | **97.30** | **97.93** | 29.60 |
| | $\alpha = 2.0$ | 85.98 | 79.10 | **93.87** | **93.88** | 93.93 | 97.96 | 51.20 | 60.57 | **70.38** | 96.92 | 97.69 | 17.60 |
| **(b) Inference Steps** | | | | | | | | | | | | | |
| — | 2 steps | 79.92 | 67.97 | 91.87 | 77.18 | 92.90 | 98.18 | 16.80 | 51.69 | 55.44 | 94.20 | 97.55 | 30.40 |
| | 5 steps | 86.03 | 77.59 | 94.48 | 88.53 | 95.65 | **98.25** | 50.00 | 60.21 | 59.96 | 97.03 | 99.25 | 29.20 |
| | 10 steps | 87.69 | 80.55 | **94.83** | 91.39 | **96.02** | 98.10 | 66.40 | 62.42 | 68.57 | **97.78** | **99.33** | 26.40 |
| | 20 steps | **87.84** | **81.17** | 94.51 | **89.54** | 95.89 | 98.10 | **79.88** | 61.02 | **68.99** | 97.07 | 99.10 | **31.10** |
| **(c) Training Iterations** | | | | | | | | | | | | | |
| 150 | $\alpha = 1.0$ | 78.23 | 73.36 | 83.11 | 80.29 | 86.09 | 98.63 | 50.80 | **60.19** | **63.62** | 88.67 | 88.47 | **34.00** |
| | $\alpha = 1.4$ | 79.33 | 73.01 | 85.65 | 81.61 | 86.20 | 98.74 | 48.00 | 59.32 | 61.92 | 90.79 | 91.11 | 26.91 |
| | $\alpha = 1.7$ | 82.79 | 74.99 | 90.60 | 84.37 | 89.92 | 98.84 | 48.40 | 60.05 | 63.25 | 93.62 | 95.99 | 31.60 |
| | $\alpha = 2.0$ | **84.31** | **76.25** | **92.37** | **87.23** | **91.86** | **98.86** | 53.97 | 59.03 | 62.36 | **95.71** | **97.04** | 30.20 |
| 450 | $\alpha = 1.0$ | 79.27 | 74.20 | 84.35 | 82.20 | 86.81 | **98.32** | 59.20 | 59.48 | 62.62 | 90.32 | 89.00 | **33.60** |
| | $\alpha = 1.4$ | 85.72 | 79.41 | 92.04 | 88.07 | 93.34 | 98.12 | **73.54** | 61.88 | 66.69 | 94.94 | 96.98 | 32.94 |
| | $\alpha = 1.7$ | **87.37** | **80.95** | 93.80 | 91.67 | **95.51** | 97.61 | 72.54 | **62.25** | 69.91 | **96.87** | **98.23** | 30.34 |
| | $\alpha = 2.0$ | 85.96 | 79.54 | 92.39 | 92.88 | 94.12 | 97.46 | 60.69 | 61.04 | **70.22** | 96.51 | 97.52 | 14.51 |
| 750 | $\alpha = 1.0$ | 80.17 | 74.35 | 85.99 | 83.16 | 86.99 | **98.05** | 62.40 | 58.44 | 62.49 | 91.23 | 90.70 | 34.40 |
| | $\alpha = 1.4$ | 86.46 | 79.79 | 93.13 | 88.64 | 93.54 | 97.68 | **78.80** | 61.12 | 67.47 | 95.72 | 97.60 | **38.40** |
| | $\alpha = 1.7$ | **87.11** | **80.42** | 93.80 | 92.93 | **95.58** | 97.72 | 62.80 | **61.78** | 70.34 | **97.30** | **97.93** | 29.60 |
| | $\alpha = 2.0$ | 85.98 | 79.10 | **93.87** | **93.88** | 93.93 | 97.96 | 51.20 | 60.57 | **70.38** | 96.92 | 97.69 | 17.60 |
| 900 | $\alpha = 1.0$ | 81.78 | 74.63 | 88.93 | 84.23 | 88.48 | 98.44 | 60.73 | 58.56 | 60.10 | 93.14 | 93.73 | **32.80** |
| | $\alpha = 1.4$ | **87.69** | **80.55** | **94.83** | 91.39 | **96.02** | 98.10 | **66.40** | **62.42** | 68.57 | **97.78** | **99.33** | 26.40 |
| | $\alpha = 1.7$ | 86.93 | 79.19 | 94.67 | 94.78 | 95.80 | 98.22 | 41.20 | 61.84 | 70.14 | 98.31 | 99.26 | 18.00 |
| | $\alpha = 2.0$ | 85.91 | 77.96 | 93.86 | **95.15** | 94.26 | **98.38** | 32.40 | 60.57 | 70.17 | 98.22 | 98.95 | 6.15 |
| 1200 | $\alpha = 1.0$ | 82.08 | 75.01 | 89.14 | 84.81 | 88.30 | 98.15 | 63.20 | 58.17 | 61.90 | 93.02 | 94.04 | **34.40** |
| | $\alpha = 1.4$ | **87.32** | **80.30** | **94.34** | 90.72 | **95.44** | 97.58 | **73.20** | **61.51** | 68.01 | **97.19** | **98.83** | 29.89 |
| | $\alpha = 1.7$ | 86.86 | 79.70 | 94.02 | **93.66** | 95.46 | 98.02 | 54.80 | 60.77 | **69.70** | 97.65 | 98.77 | 18.80 |
| | $\alpha = 2.0$ | 86.01 | 78.32 | 93.69 | 94.21 | 94.16 | **98.40** | 41.20 | 59.51 | 69.96 | 97.90 | 98.74 | 9.16 |

## C  COMPREHENSIVE QUANTITATIVE RESULTS

### C.1  HYPERPARAMETER STUDY

To validate our hyperparameter choices and understand their impact on performance, we conducted a series of studies, summarized in Table 4.

**Lora Configurations** Lora rank is a critical ingredient that influences the fine-tuning performance. As we know, Lora learns much less with small ranks compared to full fine-tuning Biderman et al. (2024), thus, Lora rank should be large enough to have the capacity to learn the new capabilities since tasks like I2V are very general. We investigated the influence of LoRA rank, a proxy for the adaptation's capacity. As shown in Table 4(a), a higher rank of 512 consistently outperforms a rank of 256 on most metrics, particularly in overall quality. This suggests that a larger adaptation capacity is beneficial in capturing the nuances of temporal dynamics required for I2V tasks. We also find that the LoRA alpha scaling at inference time is critical; an alpha of 1.7 yields the best results for the 750-iteration checkpoint, balancing the influence of the LoRA weights against the pre-trained model.

**Inference Steps.** As detailed in Table 4(b), we analyzed the trade-off between computational cost at inference and generation quality using our best checkpoint with rank 512 and alpha 1.4 with 900 iterations. Performance scales predictably with the number of steps, with significant gains observed up to 10 steps. While 20 steps provide a marginal improvement, the results at 10 steps are nearly identical (87.69 vs. 87.84). Consequently, we adopt 10 inference steps as our default to ensure an optimal balance between quality and generation speed.

**Training Progression.** We evaluated checkpoints of Lora rank 512 at various stages of training, from 150 to 1200 iterations, using 10 inference steps. Table 4(c) shows a clear trend of improving performance up to 900 iterations, where we achieved our highest score of 87.69 with an alpha of 1.4. Beyond this point, performance begins to plateau or slightly degrade, indicating that the model has converged. This rapid convergence underscores the data efficiency of our approach. Our final model for comparison in Table 1 uses the 900-iteration checkpoint of rank 512.

## C.2 ABLATION STUDIES

We conduct further ablation studies to analyze the core components of our proposed method. We investigate the effectiveness of our training strategy against a baseline approach and examine the impact of different timestep sampling strategies during training. The results are summarized in Table 5.

**Training Strategy.** We compare our training strategy against the baseline method used in Wan-I2V across different training regimes: full model fine-tuning and parameter-efficient LoRA fine-tuning. As shown in Table 5(a), both of our fine-tuning approaches significantly outperform the baseline at every checkpoint. Our full fine-tuning method already achieves a total score of 83.78 in just 150 steps, surpassing the baseline's peak score of 75.01. The LoRA-based approach further accelerates performance gains. The results show that performance consistently improves with more training steps. For instance, with $\alpha = 1.7$, the score increases from 84.92 at 150 steps to 86.27 at 600 steps. The best LoRA result is 87.09 at 900 steps with $\alpha = 1.4$, which substantially outperforms full fine-tuning at the same iteration (84.06). This highlights not only the effectiveness of our training recipe but also the remarkable efficiency and power of combining it with LoRA.

**Timestep Sampling Strategy.** Our method utilizes a simple yet effective strategy of sampling purely random timesteps for each frame during training. To validate this choice, we compare it against an intuitive baseline for I2V with the first frame always being noise-free $\tau^1 = 0$ while all other timesteps are of the same value, and a more structured approach, PTSS, where a certain percentage of timesteps are sampled from a specific range. As detailed in Table 5(b), we conduct a comprehensive comparison at the 900-iteration mark, testing PTSS with probabilities ($p$) of 0.2, 0.5, and 0.8 against our fully random approach (equivalent to $p = 0$). The results consistently show that our method outperforms I2V beaseline and all PTSS variants across different LoRA alpha values. Notably, our approach with $\alpha = 1.4$ achieves the highest total score of 87.69, surpassing the best PTSS result (87.36 from p=0.5, $\alpha = 1.7$). This confirms that a simpler, fully randomized timestep sampling strategy is more effective for learning robust temporal dynamics in our framework.

Table 5: **Ablation studies on training methodology.** This table presents our analysis of (a) the training strategy, comparing our method (with both Full and LoRA fine-tuning) against the baseline across different training steps, and (b) the timestep sampling strategy. All scores are reported in percentages (%), with higher values being better. The overall best configuration is marked with *.

| Group | Setting | Overall | | | Quality Metrics | | | | | | I2V Metrics | | |
|---|---|---|---|---|---|---|---|---|---|---|---|---|---|
| | | Total | Quality | I2V | SC | BC | MS | DD | AQ | IQ | I2V-S | I2V-B | CM |
| **(a) Training Strategy** | | | | | | | | | | | | | |
| Baseline (Wan-I2V) | 150 steps | 75.01 | 77.85 | 72.16 | 97.02 | 98.31 | 99.22 | 24.80 | 59.75 | 63.65 | 82.19 | 77.43 | 29.60 |
| | 300 steps | 74.02 | 76.81 | 71.23 | 96.93 | 97.75 | 99.18 | 18.80 | 58.28 | 63.39 | 81.58 | 76.48 | 30.00 |
| | 600 steps | 73.29 | 75.63 | 70.95 | 97.44 | 97.88 | 99.23 | 9.60 | 56.63 | 62.18 | 81.47 | 76.09 | 30.80 |
| | 900 steps | 73.21 | 76.02 | 70.39 | 96.96 | 97.65 | 99.13 | 17.20 | 57.00 | 61.42 | 80.56 | 76.15 | 28.80 |
| | 1500 steps | 73.39 | 76.26 | 70.53 | 97.19 | 98.00 | 99.30 | 11.20 | 59.24 | 62.11 | 80.60 | 76.13 | 31.60 |
| Ours (Full Finetune) | 150 steps | 83.78 | 76.10 | 91.46 | 85.20 | 91.10 | 98.79 | 39.60 | 63.46 | 67.95 | 94.62 | 96.14 | 36.00 |
| | 300 steps | 83.35 | 78.07 | 88.62 | 84.86 | 90.91 | 98.37 | 66.40 | 63.61 | 67.35 | 92.77 | 93.57 | 32.80 |
| | 600 steps | 85.21 | 78.51 | 91.90 | 85.95 | 90.20 | 98.30 | 76.00 | 61.80 | 66.69 | 95.58 | 95.79 | 38.80 |
| | 900 steps | 84.06 | 77.67 | 90.45 | 85.97 | 90.67 | 98.40 | 66.80 | 61.27 | 66.17 | 94.27 | 95.05 | 33.60 |
| Ours (LoRA Finetune) | 150 steps, $\alpha = 1.4$ | 81.62 | 74.62 | 88.62 | 83.56 | 88.10 | 98.90 | 45.60 | 60.52 | 65.39 | 92.20 | 94.30 | 29.60 |
| | 150 steps, $\alpha = 1.7$ | 84.92 | 77.39 | 92.44 | 87.18 | 93.12 | 99.02 | 46.00 | 62.73 | 66.76 | 95.52 | 97.08 | 33.20 |
| | 300 steps, $\alpha = 1.4$ | 82.97 | 75.39 | 90.55 | 86.42 | 90.99 | 98.94 | 40.80 | 59.90 | 65.19 | 93.87 | 95.67 | 32.00 |
| | 600 steps, $\alpha = 1.4$ | 85.98 | 78.60 | 93.35 | 89.37 | 93.50 | 98.76 | 58.80 | 60.55 | 66.98 | 96.49 | 97.87 | 30.40 |
| | 600 steps, $\alpha = 1.7$ | 86.27 | 78.73 | 93.81 | 91.78 | 94.51 | 98.40 | 52.40 | 60.13 | 68.37 | 97.31 | 98.43 | 22.80 |
| | 900 steps, $\alpha = 1.4$* | 87.09 | 79.77 | 94.42 | 90.37 | 94.68 | 98.55 | 62.80 | 61.72 | 68.20 | 97.40 | 98.67 | 31.20 |
| | 900 steps, $\alpha = 1.7$ | 87.01 | 80.02 | 94.00 | 93.56 | 96.01 | 98.44 | 51.20 | 61.74 | 70.24 | 97.70 | 98.54 | 20.80 |
| **(b) Timestep Sampling Strategy** | | | | | | | | | | | | | |
| Ours | 450 steps, $\alpha = 1.0$ | 79.27 | 74.20 | 84.35 | 82.20 | 86.81 | 98.32 | 59.20 | 59.48 | 62.62 | 90.32 | 89.00 | 33.60 |
| | 450 steps, $\alpha = 1.4$ | 85.72 | 79.41 | 92.04 | 88.07 | 93.34 | 98.12 | 73.54 | 61.88 | 66.69 | 94.94 | 96.98 | 32.94 |
| | 450 steps, $\alpha = 1.7$ | 87.37 | 80.95 | 93.80 | 91.67 | 95.51 | 97.61 | 72.54 | 62.25 | 69.91 | 96.87 | 98.23 | 30.34 |
| | 900 steps, $\alpha = 1.0$ | 81.78 | 74.63 | 88.93 | 84.23 | 88.48 | 98.44 | 60.73 | 58.56 | 60.10 | 93.14 | 93.73 | 32.80 |
| | 900 steps, $\alpha = 1.4$* | 87.69 | 80.55 | 94.83 | 91.39 | 96.02 | 98.10 | 66.40 | 62.42 | 68.57 | 97.78 | 99.33 | 26.40 |
| | 900 steps, $\alpha = 1.7$ | 86.93 | 79.19 | 94.67 | 94.78 | 95.80 | 98.22 | 41.20 | 61.84 | 70.14 | 98.31 | 99.26 | 18.00 |
| I2V ($\tau^1 = 0$) | 450 steps, $\alpha = 1.0$ | 74.26 | 71.62 | 76.91 | 76.83 | 84.03 | 98.41 | 54.40 | 58.14 | 61.92 | 82.44 | 84.49 | 30.80 |
| | 450 steps, $\alpha = 1.4$ | 74.03 | 71.11 | 76.95 | 76.55 | 83.58 | 98.49 | 53.60 | 56.12 | 62.20 | 82.48 | 84.61 | 29.60 |
| | 450 steps, $\alpha = 1.7$ | 74.73 | 70.98 | 78.47 | 76.63 | 83.91 | 98.56 | 49.60 | 56.23 | 62.63 | 83.65 | 85.96 | 29.60 |
| | 900 steps, $\alpha = 1.0$ | 73.27 | 69.96 | 76.57 | 75.93 | 83.37 | 98.60 | 43.20 | 56.57 | 61.25 | 82.23 | 84.15 | 30.80 |
| | 900 steps, $\alpha = 1.4$ | 72.54 | 68.47 | 76.61 | 75.11 | 83.10 | 98.68 | 36.40 | 53.87 | 60.23 | 81.61 | 84.33 | 36.40 |
| | 900 steps, $\alpha = 1.7$ | 72.42 | 67.82 | 77.02 | 74.53 | 82.96 | 98.72 | 32.80 | 53.39 | 59.66 | 81.90 | 85.05 | 32.00 |
| PTSS ($p = 0.2$) | 450 steps, $\alpha = 1.0$ | 77.53 | 72.33 | 82.74 | 79.62 | 85.66 | 98.61 | 42.40 | 59.96 | 63.85 | 88.18 | 88.80 | 27.60 |
| | 450 steps, $\alpha = 1.4$ | 79.67 | 73.63 | 85.70 | 81.07 | 87.44 | 98.69 | 46.40 | 60.23 | 64.35 | 90.77 | 90.77 | 32.80 |
| | 450 steps, $\alpha = 1.7$ | 82.05 | 75.32 | 88.79 | 82.47 | 89.80 | 98.76 | 50.80 | 62.16 | 64.42 | 92.70 | 94.19 | 28.80 |
| | 900 steps, $\alpha = 1.0$ | 77.96 | 72.24 | 83.68 | 80.38 | 85.49 | 98.40 | 44.40 | 58.67 | 63.72 | 89.43 | 88.91 | 31.20 |
| | 900 steps, $\alpha = 1.4$ | 81.57 | 74.71 | 88.43 | 83.63 | 88.95 | 98.51 | 52.40 | 59.57 | 63.50 | 92.69 | 93.51 | 30.40 |
| | 900 steps, $\alpha = 1.7$ | 84.74 | 77.60 | 91.88 | 85.97 | 92.68 | 98.40 | 63.20 | 61.27 | 64.93 | 94.68 | 96.82 | 34.80 |
| PTSS ($p = 0.5$) | 450 steps, $\alpha = 1.0$ | 78.16 | 72.96 | 83.36 | 80.69 | 86.58 | 98.42 | 45.20 | 59.80 | 64.21 | 88.97 | 88.64 | 33.60 |
| | 450 steps, $\alpha = 1.4$ | 82.08 | 75.76 | 88.41 | 84.98 | 91.24 | 98.46 | 48.40 | 61.64 | 64.70 | 91.94 | 94.20 | 29.60 |
| | 450 steps, $\alpha = 1.7$ | 85.52 | 78.64 | 92.41 | 88.87 | 94.94 | 98.45 | 52.40 | 63.29 | 67.36 | 95.06 | 97.40 | 33.60 |
| | 900 steps, $\alpha = 1.0$ | 79.63 | 73.51 | 85.76 | 82.98 | 87.27 | 98.37 | 46.00 | 58.80 | 64.42 | 91.01 | 90.80 | 30.67 |
| | 900 steps, $\alpha = 1.4$ | 86.38 | 79.51 | 93.26 | 90.18 | 95.60 | 98.26 | 56.30 | 63.74 | 68.01 | 96.20 | 98.09 | 28.75 |
| | 900 steps, $\alpha = 1.7$ | 87.36 | 80.44 | 94.28 | 93.25 | 96.63 | 98.07 | 54.80 | 63.58 | 69.67 | 97.71 | 98.36 | 28.80 |
| PTSS ($p = 0.8$) | 450 steps, $\alpha = 1.0$ | 79.03 | 73.78 | 84.28 | 82.13 | 86.86 | 98.56 | 45.60 | 60.41 | 65.36 | 89.98 | 89.14 | 34.40 |
| | 450 steps, $\alpha = 1.4$ | 85.59 | 79.03 | 92.14 | 88.10 | 93.93 | 98.63 | 61.60 | 62.84 | 67.03 | 94.78 | 97.35 | 32.00 |
| | 450 steps, $\alpha = 1.7$ | 87.31 | 80.84 | 93.79 | 91.63 | 95.98 | 98.17 | 65.20 | 63.27 | 69.41 | 97.02 | 98.08 | 30.40 |
| | 900 steps, $\alpha = 1.0$ | 80.71 | 75.17 | 86.25 | 82.65 | 86.73 | 97.73 | 69.60 | 59.37 | 64.50 | 91.37 | 90.96 | 34.80 |
| | 900 steps, $\alpha = 1.4$ | 85.98 | 78.86 | 93.10 | 87.10 | 91.34 | 97.31 | 86.40 | 60.79 | 64.98 | 96.13 | 97.24 | 38.00 |
| | 900 steps, $\alpha = 1.7$ | 86.49 | 79.30 | 93.69 | 89.98 | 92.78 | 97.11 | 79.20 | 60.44 | 66.67 | 97.15 | 97.67 | 32.40 |

# D  QUALITATIVE RESULTS ON MULTI-TASK CAPABILITIES

Our approach demonstrates exceptional generalization across diverse video generation tasks beyond text-to-video (T2V), without task-specific training. This capability comes from flexible vectorized timestep settings, enabling arbitrary conditioning on any subset of frames.

**Text-to-Video Generation.**  Unlike specialized image-to-video (I2V) models, Pusa preserves the T2V capabilities of its foundation model. As shown in Fig. 14, the qualitative output maintains high quality, demonstrating that our fine-tuning process does not induce catastrophic forgetting of the primary T2V task. This preservation of capabilities establishes Pusa as a truly unified video generation model.

**Complex Temporal Tasks.**  The FVDM framework reveals its true power through zero-shot performance on complex temporal synthesis tasks. Additional results demonstrating seamless I2V generation are presented in Fig. 4.

Pusa can perform start-end frames conditioning through various configurations. When conditioning on the first frame and the last frame (encoded to a single latent frame similar to the first frame), the model generates coherent video sequences, as illustrated in Fig. 8. Alternatively, conditioning on the first frame and the last 4 frames (encoded to a single latent frame as default) yields improved results, as shown in Fig. 10. The latter approach addresses the challenge posed by the $4\times$ compression rate for the last frames introduced by the VAE, where conditioning solely on the last frame produces inferior results due to its interpretation as 4 static frames.

Using the unique properties of our framework, we can enhance video coherence by introducing controlled noise to the encoded latents (e.g., setting $\tau^1 = 0.3 * t$ and $\tau^N = 0.7 * t$). This technique generates appropriate motion and content for the condition frames, resulting in more coherent video synthesis, as demonstrated in Fig. 9.

Furthermore, Figures 11, 12, and 13 showcase Pusa's capabilities for video completion/transition and video extension, seamlessly completing or continuing given video sequences. These advanced capabilities emerge inherently from our vectorized timestep adaptation strategy, which does not require task-specific training, and highlight the remarkable versatility and power of our approach.

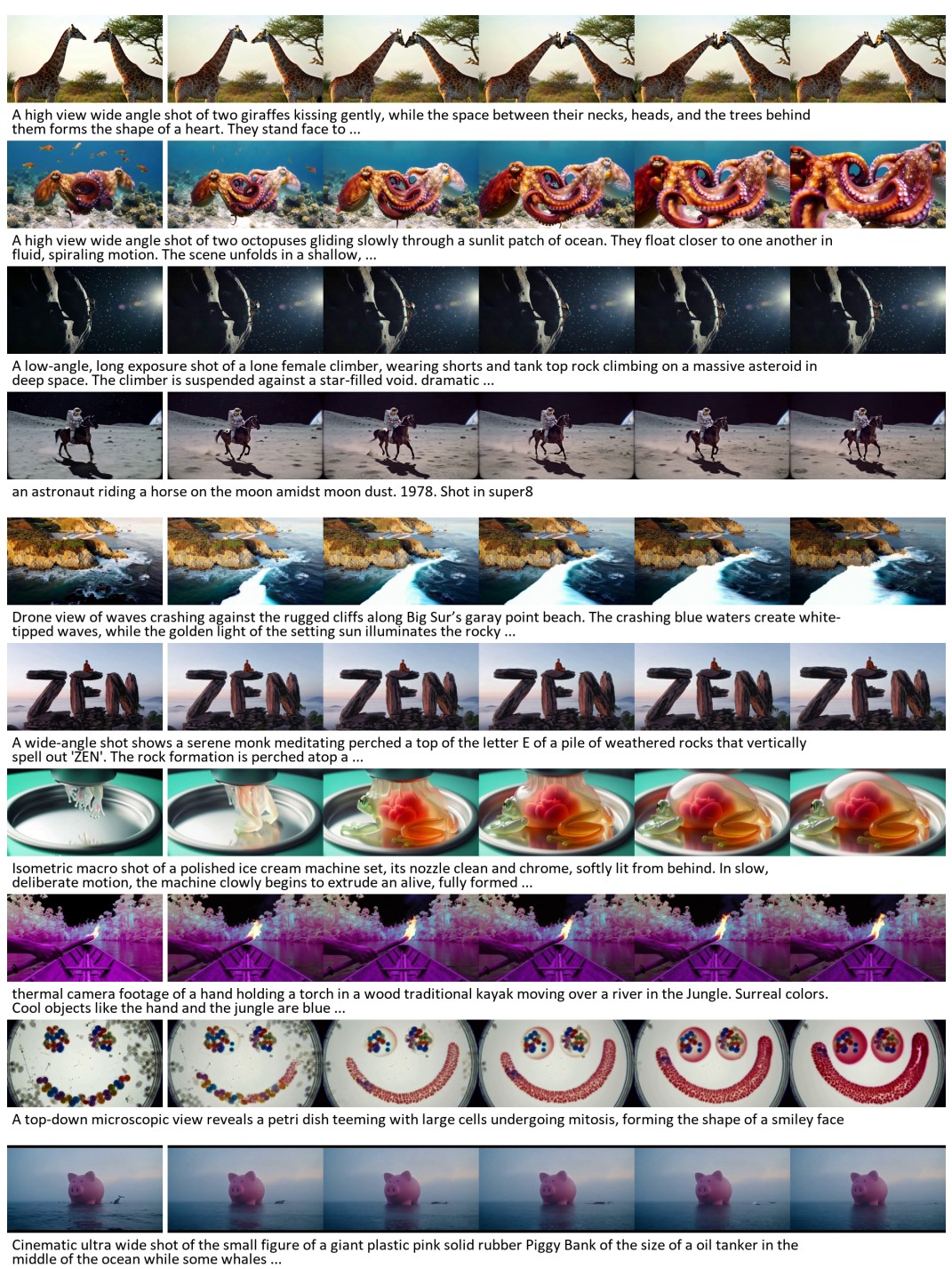

A high view wide angle shot of two giraffes kissing gently, while the space between their necks, heads, and the trees behind them forms the shape of a heart. They stand face to ...

A high view wide angle shot of two octopuses gliding slowly through a sunlit patch of ocean. They float closer to one another in fluid, spiraling motion. The scene unfolds in a shallow, ...

A low-angle, long exposure shot of a lone female climber, wearing shorts and tank top rock climbing on a massive asteroid in deep space. The climber is suspended against a star-filled void. dramatic ...

an astronaut riding a horse on the moon amidst moon dust. 1978. Shot in super8

Drone view of waves crashing against the rugged cliffs along Big Sur's garay point beach. The crashing blue waters create white-tipped waves, while the golden light of the setting sun illuminates the rocky ...

A wide-angle shot shows a serene monk meditating perched a top of the letter E of a pile of weathered rocks that vertically spell out 'ZEN'. The rock formation is perched atop a ...

Isometric macro shot of a polished ice cream machine set, its nozzle clean and chrome, softly lit from behind. In slow, deliberate motion, the machine clowly begins to extrude an alive, fully formed ...

thermal camera footage of a hand holding a torch in a wood traditional kayak moving over a river in the Jungle. Surreal colors. Cool objects like the hand and the jungle are blue ...

A top-down microscopic view reveals a petri dish teeming with large cells undergoing mitosis, forming the shape of a smiley face

Cinematic ultra wide shot of the small figure of a giant plastic pink solid rubber Piggy Bank of the size of a oil tanker in the middle of the ocean while some whales ...

Figure 7: More image-to-video results. The first frames of each row are the given condition images extracted from Veo2 & Sora demos. Each generated video has 81 frames in total.

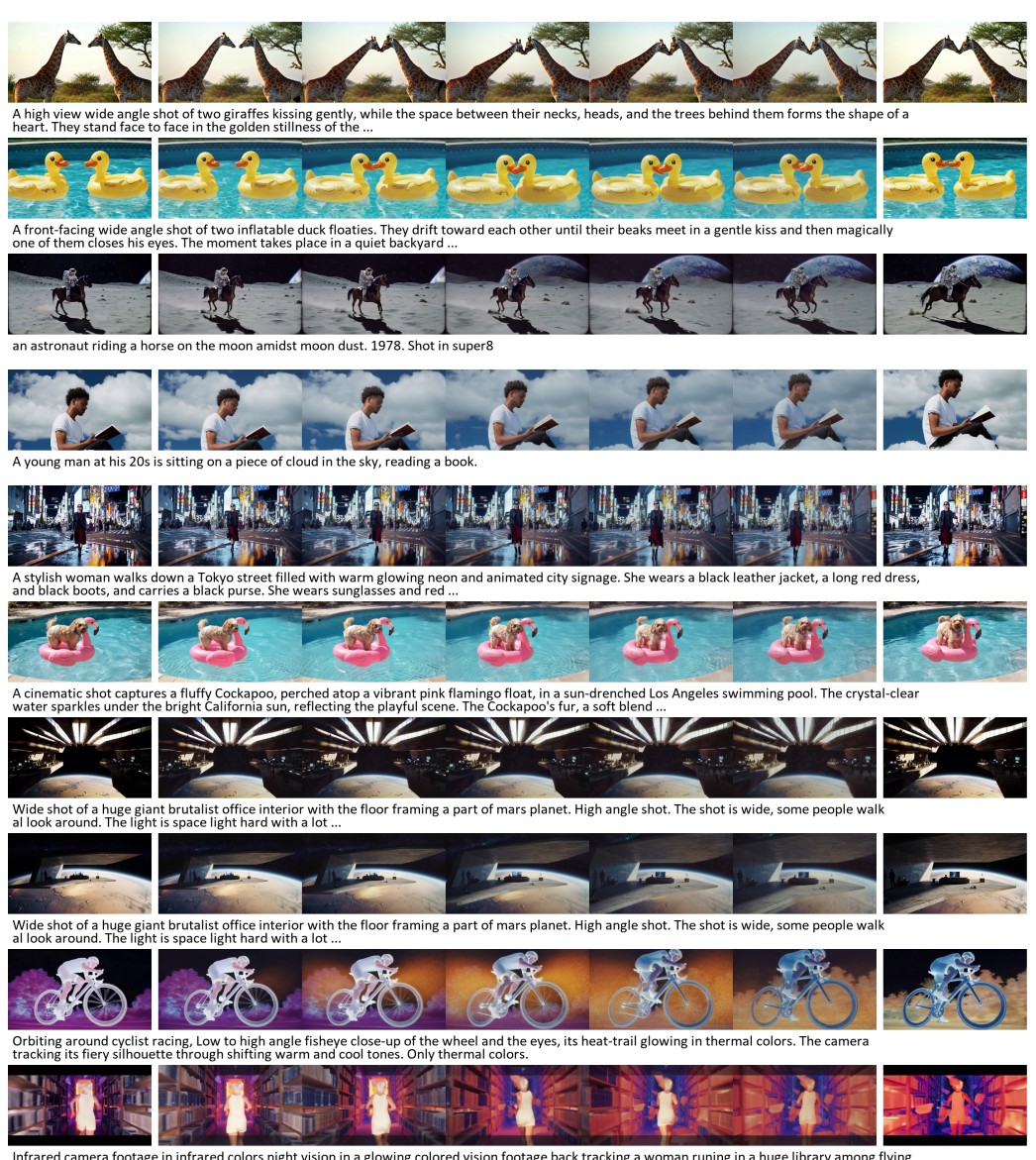

A high view wide angle shot of two giraffes kissing gently, while the space between their necks, heads, and the trees behind them forms the shape of a heart. They stand face to face in the golden stillness of the ...

A front-facing wide angle shot of two inflatable duck floaties. They drift toward each other until their beaks meet in a gentle kiss and then magically one of them closes his eyes. The moment takes place in a quiet backyard ...

an astronaut riding a horse on the moon amidst moon dust. 1978. Shot in super8

A young man at his 20s is sitting on a piece of cloud in the sky, reading a book.

A stylish woman walks down a Tokyo street filled with warm glowing neon and animated city signage. She wears a black leather jacket, a long red dress, and black boots, and carries a black purse. She wears sunglasses and red ...

A cinematic shot captures a fluffy Cockapoo, perched atop a vibrant pink flamingo float, in a sun-drenched Los Angeles swimming pool. The crystal-clear water sparkles under the bright California sun, reflecting the playful scene. The Cockapoo's fur, a soft blend ...

Wide shot of a huge giant brutalist office interior with the floor framing a part of mars planet. High angle shot. The shot is wide, some people walk al look around. The light is space light hard with a lot ...

Wide shot of a huge giant brutalist office interior with the floor framing a part of mars planet. High angle shot. The shot is wide, some people walk al look around. The light is space light hard with a lot ...

Orbiting around cyclist racing, Low to high angle fisheye close-up of the wheel and the eyes, its heat-trail glowing in thermal colors. The camera tracking its fiery silhouette through shifting warm and cool tones. Only thermal colors.

Infrared camera footage in infrared colors night vision in a glowing colored vision footage back tracking a woman runing in a huge library among flying papers, 50mm lens. Wide lens. narrow light spectrum. Unnatural colors

Figure 8: Zero-shot results w.r.t. start & end frames to video. The first and last frames are given condition frames extracted from Veo2 & Sora demos. Each generated video has 81 frames in total.

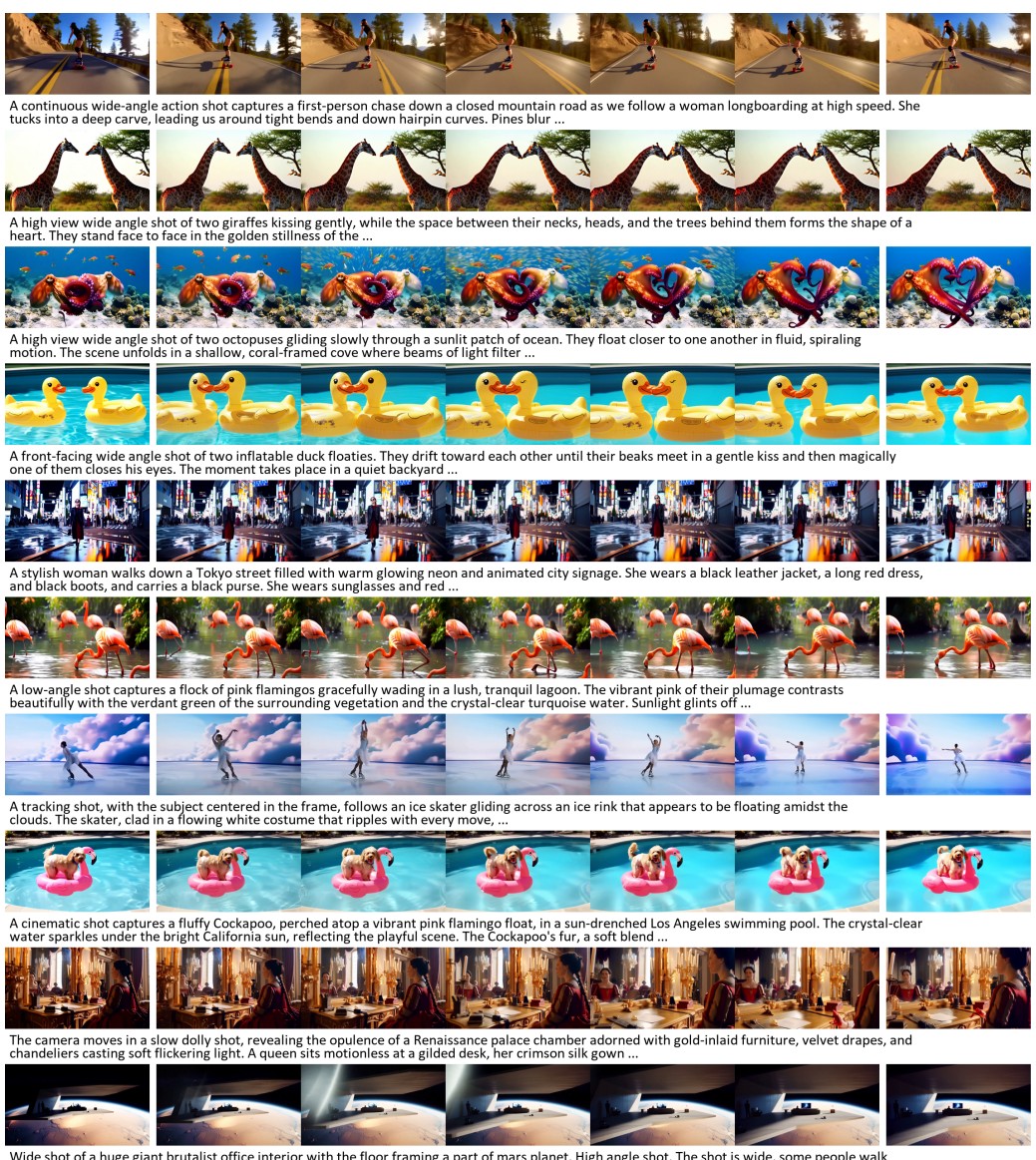

Figure 9: Zero-shot results w.r.t. start & end frames with noise. The first and last frames are given conditions and added 30% and 70% noise during sampling to make the generated video more coherent. Condition frames are extracted from Veo2 & Sora demos. Each generated video has 81 frames in total.

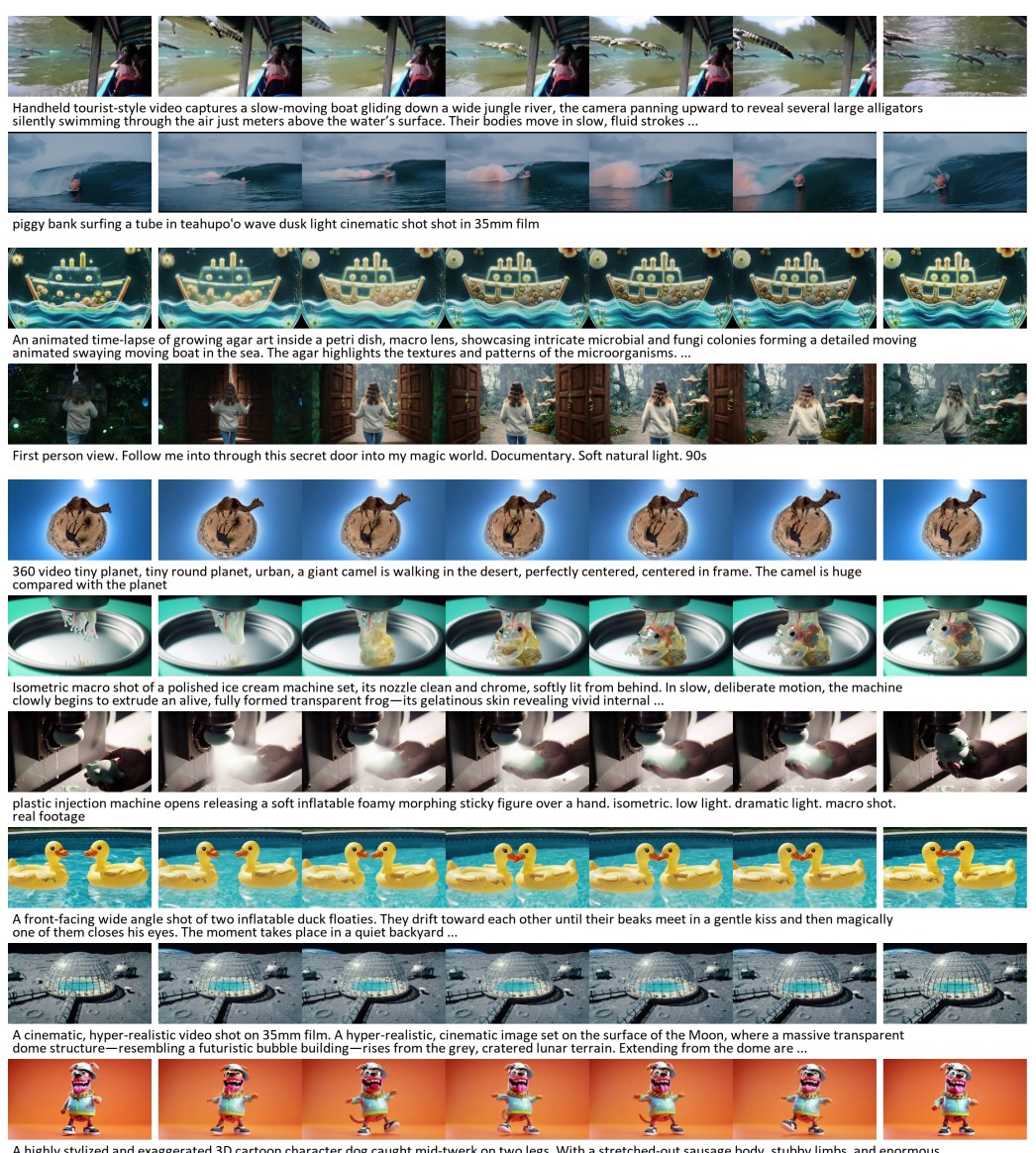

Handheld tourist-style video captures a slow-moving boat gliding down a wide jungle river, the camera panning upward to reveal several large alligators silently swimming through the air just meters above the water's surface. Their bodies move in slow, fluid strokes ...

piggy bank surfing a tube in teahupo'o wave dusk light cinematic shot shot in 35mm film

An animated time-lapse of growing agar art inside a petri dish, macro lens, showcasing intricate microbial and fungi colonies forming a detailed moving animated swaying moving boat in the sea. The agar highlights the textures and patterns of the microorganisms. ...

First person view. Follow me into through this secret door into my magic world. Documentary. Soft natural light. 90s

360 video tiny planet, tiny round planet, urban, a giant camel is walking in the desert, perfectly centered, centered in frame. The camel is huge compared with the planet

Isometric macro shot of a polished ice cream machine set, its nozzle clean and chrome, softly lit from behind. In slow, deliberate motion, the machine clowly begins to extrude an alive, fully formed transparent frog—its gelatinous skin revealing vivid internal ...

plastic injection machine opens releasing a soft inflatable foamy morphing sticky figure over a hand. isometric. low light. dramatic light. macro shot. real footage

A front-facing wide angle shot of two inflatable duck floaties. They drift toward each other until their beaks meet in a gentle kiss and then magically one of them closes his eyes. The moment takes place in a quiet backyard ...

A cinematic, hyper-realistic video shot on 35mm film. A hyper-realistic, cinematic image set on the surface of the Moon, where a massive transparent dome structure—resembling a futuristic bubble building—rises from the grey, cratered lunar terrain. Extending from the dome are ...

A highly stylized and exaggerated 3D cartoon character dog caught mid-twerk on two legs. With a stretched-out sausage body, stubby limbs, and enormous cartoon eyes, he wears a shiny silver tracksuit with baby blue and neon green stripes. His mouth ...

Figure 10: Zero-shot results w.r.t. start & end frames to video. The first and last 4 frames (encoded to one latent frame) are given condition frames extracted from Veo2 & Sora demos. Each generated video has 81 frames/21 latent frame in total.

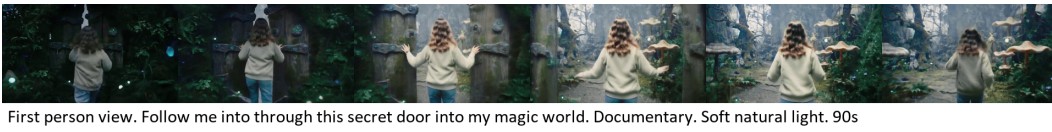

First person view. Follow me into through this secret door into my magic world. Documentary. Soft natural light. 90s

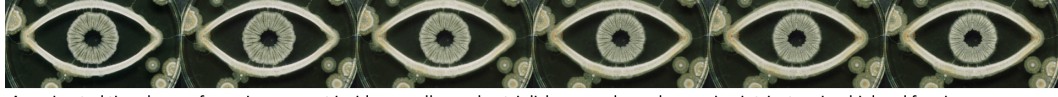

An animated time-lapse of growing agar art inside a small round petri dish, macro lens, showcasing intricate microbial and fungi colonies forming a detailed moving animated moving blinking blinking blinking moving eye. The ...

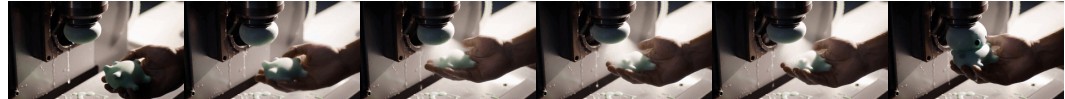

360 video tiny planet, tiny round planet, urban, a giant camel is walking in the desert, perfectly centered, centered in frame. The camel is huge compared with the planet

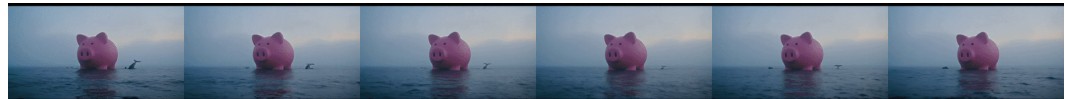

plastic injection machine opens releasing a soft inflatable foamy morphing sticky figure over a hand. isometric. low light. dramatic light. macro shot. real footage

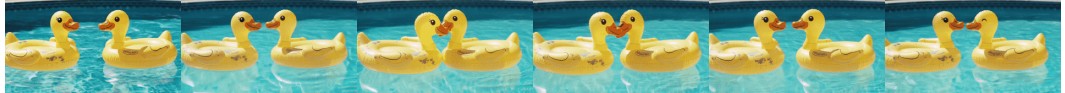

Cinematic ultra wide shot of the small figure of a giant plastic pink solid rubber Piggy Bank of the size of a oil tanker in the middle of the ocean while some whales ...

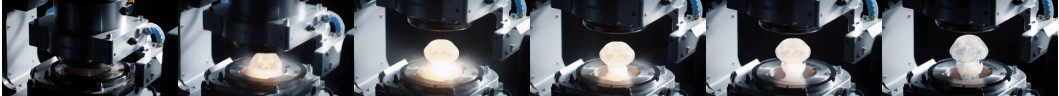

A front-facing wide angle shot of two inflatable duck floaties. They drift toward each other until their beaks meet in a gentle kiss and then magically one of them closes his eyes. The ...

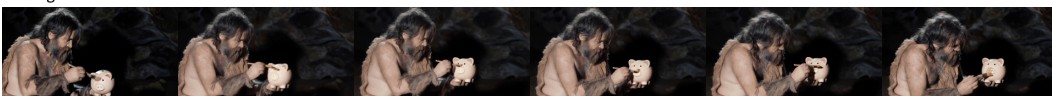

clean metal plastic injection machine opens releasing a pinkish jelly fish. isometric. low dramatic light. macro shot. real footage. Back lit

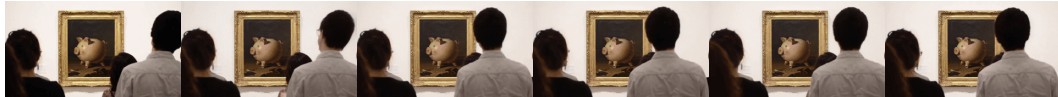

neanderthal hairy man painting a piggy bank in a cave in the dark almost fully black lit by a hand held torch, shaky hand held footage shot with a 8mm film camera, hand ...

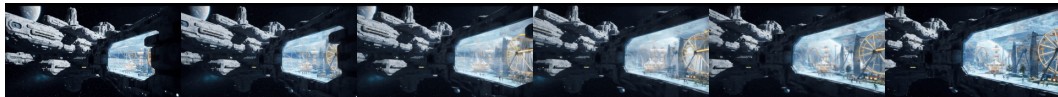

many people from behind contemplating a framed renaissance masterpiece in the wall: an oil on canvas depicting a piggy bank. Amateur video, shot with an phone

A cinematic, hyper-realistic video shot on 35mm film.A hyper-realistic, cinematic image of a massive brutalist spacecraft floating in deep space, its architecture defined by immense, angular structures made of textured concrete-like alloys—etched

Figure 11: Zero-shot results w.r.t. video completion/transition. The first 9 frames and the last 12 frames extracted from Veo2 demos are given as conditions and encoded to the first 3 latent frames and the last 3 latent frames, respectively. Each generated video has 81 frames/21 latent frames in total.

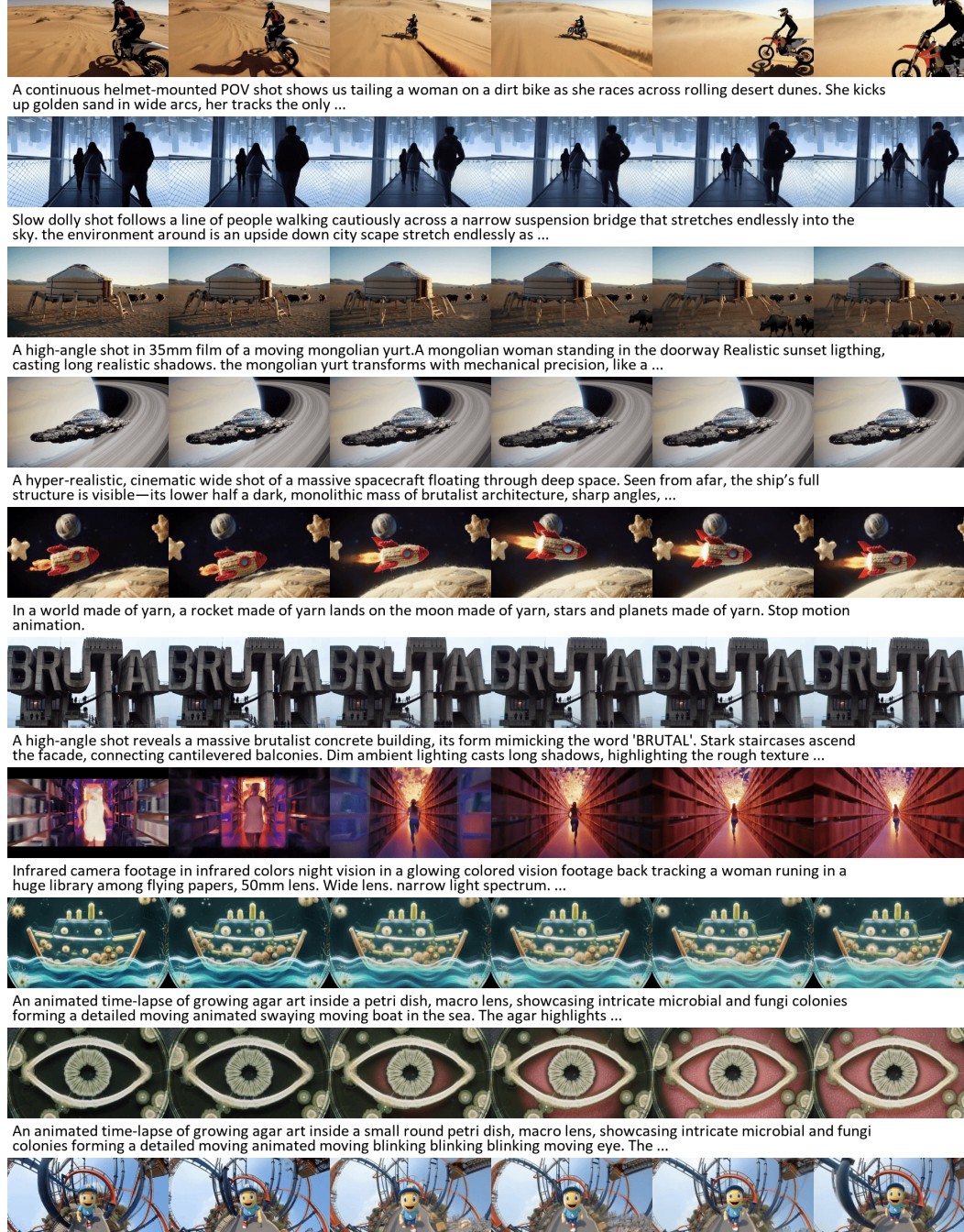

Figure 12: Zero-shot results w.r.t. video extension. The first 13 frames extracted from Veo2 demos are given as conditions and encoded to the first 4 latent frames. Each generated video has 81 frames/21 latent frames in total.

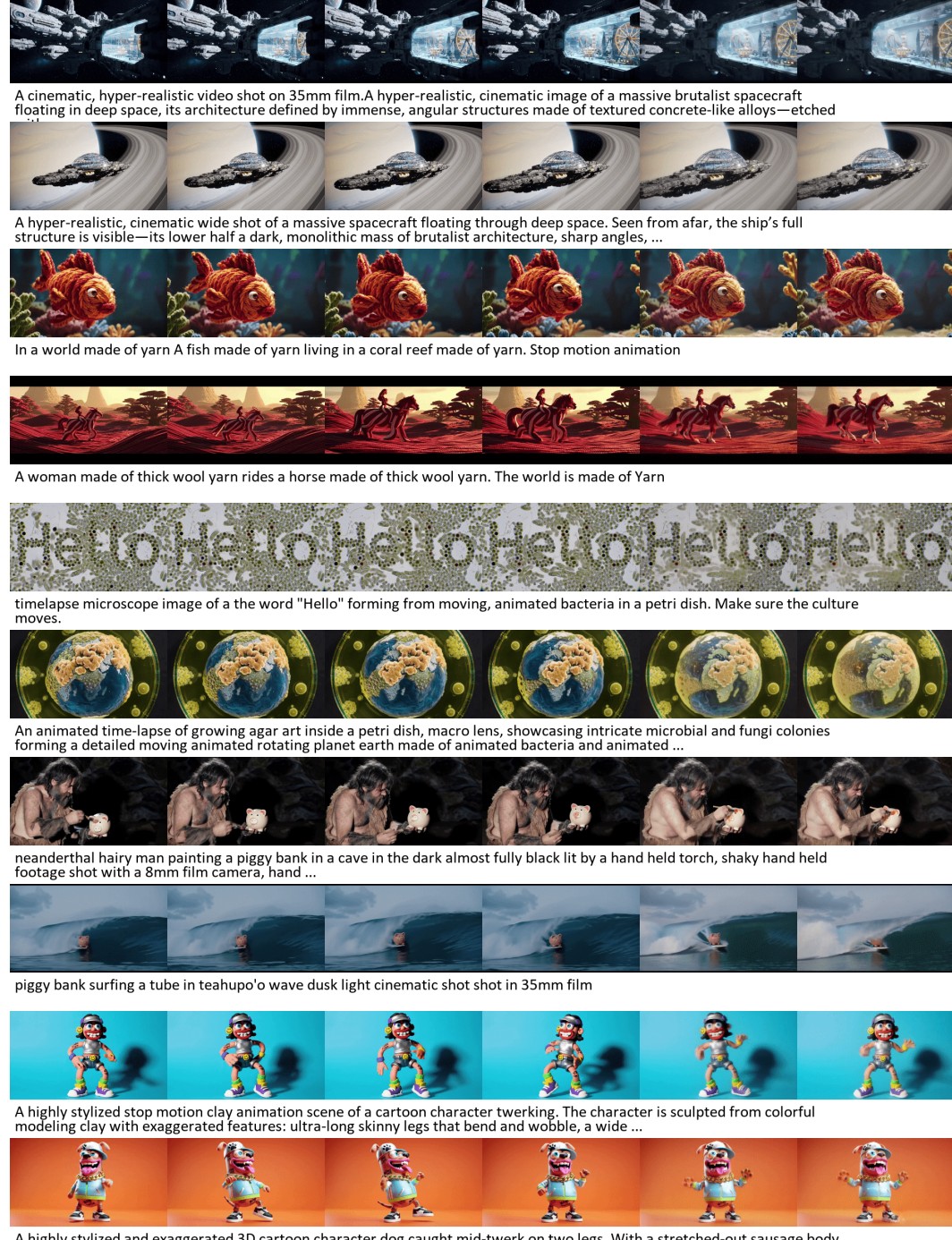

A cinematic, hyper-realistic video shot on 35mm film.A hyper-realistic, cinematic image of a massive brutalist spacecraft floating in deep space, its architecture defined by immense, angular structures made of textured concrete-like alloys—etched ...

A hyper-realistic, cinematic wide shot of a massive spacecraft floating through deep space. Seen from afar, the ship's full structure is visible—its lower half a dark, monolithic mass of brutalist architecture, sharp angles, ...

In a world made of yarn A fish made of yarn living in a coral reef made of yarn. Stop motion animation

A woman made of thick wool yarn rides a horse made of thick wool yarn. The world is made of Yarn

timelapse microscope image of a the word "Hello" forming from moving, animated bacteria in a petri dish. Make sure the culture moves.

An animated time-lapse of growing agar art inside a petri dish, macro lens, showcasing intricate microbial and fungi colonies forming a detailed moving animated rotating planet earth made of animated bacteria and animated ...

neanderthal hairy man painting a piggy bank in a cave in the dark almost fully black lit by a hand held torch, shaky hand held footage shot with a 8mm film camera, hand ...

piggy bank surfing a tube in teahupo'o wave dusk light cinematic shot shot in 35mm film

A highly stylized stop motion clay animation scene of a cartoon character twerking. The character is sculpted from colorful modeling clay with exaggerated features: ultra-long skinny legs that bend and wobble, a wide ...

A highly stylized and exaggerated 3D cartoon character dog caught mid-twerk on two legs. With a stretched-out sausage body, stubby limbs, and enormous cartoon eyes, he wears a shiny silver tracksuit with baby ...

Figure 13: Zero-shot results w.r.t. video extension. The first 41 frames extracted from Veo2 demos are given as conditions and encoded to the first 11 latent frames. Each generated video has 81 frames/21 latent frames in total.

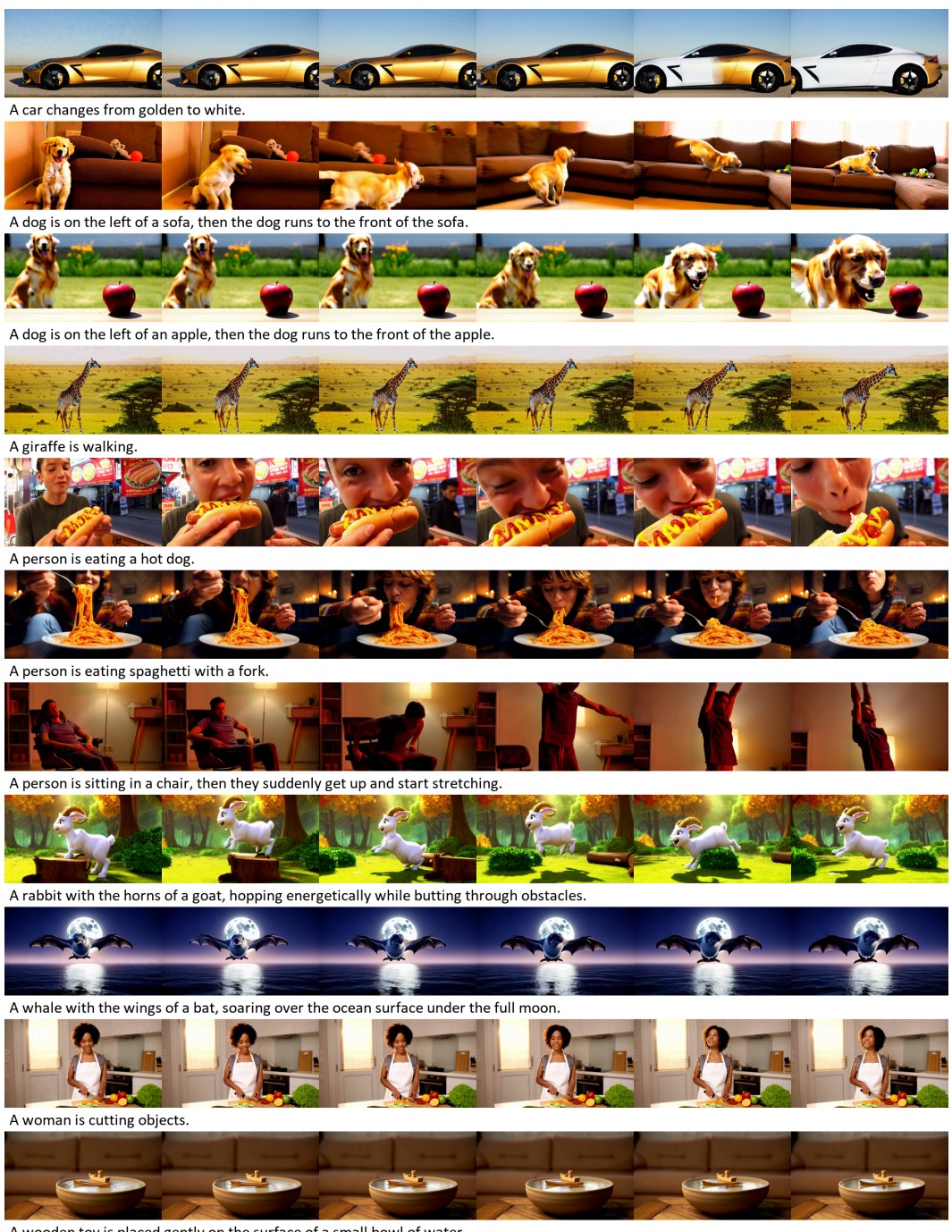

Figure 14: Text-to-video results. Prompts all from Vbench2.0.

