# OpenReview forum: "Pusa V1.0: Unlocking Temporal Control in Pretrained Video Diffusion Models via Vectorized Timestep Adaptation"
_ICLR.cc/2026/Conference — ICLR 2026 Poster_

### Official Review · Reviewer_TcSx · 2025-10-27

**Soundness:** 3
**Presentation:** 3
**Contribution:** 3
**Rating:** 8
**Confidence:** 4

**Summary:**

The paper proposes Pusa V1.0, a lightweight adaptation for large pretrained text-to-video (T2V) diffusion models that replaces the scalar timestep with a vectorized timestep (one per frame) and learns with a frame-aware flow-matching objective. The key contribution is Vectorized Timestep Adaptation, a non-destructive modification that enhance the base model’s generation ability. With minimal fine-tuning (LoRA or full FT) on ~3.9k Wan-T2V generated clips, Pusa achieves SOTA-level image-to-video (I2V) results on VBench-I2V and further shows zero-shot behavior for other downstream tasks.

**Strengths:**

1. Simple, general idea with clear motivation. Moving from a scalar to a vectorized timestep directly attacks the synchronized-frame limitation of standard VDMs; the frame-aware flow-matching formulation is clean and well explained.
2. High efficiency. Comparable I2V performance to Wan-I2V with a tiny fraction of the compute/data, plus favorable results with only 10 sampling steps. The ablations (LoRA vs full FT, timestep sampling) are helpful.
3. Unified capability. The same model handles I2V, start–end, and extension without task-specific heads or destructive finetuning.

**Weaknesses:**

1. Lack of quantitative evaluation for non-I2V tasks. While the author mentions zero-shot, start–end and other applications of its method, it lacks quantitative evaluation for these applications; most quantitative focus is I2V on VBench-I2V. Concrete metrics are missing from the main text. It would be better for the auther to add quantitative results for applications.
2. Data source and generalization. Fine-tuning uses videos generated by the same base video model. This could limit diversity and introduce bias toward the base model’s distribution; it’s unclear how well Pusa generalizes to challenging real photos as I2V inputs beyond curated benchmarks. Could you provide some details of the ~3,860 Wan-T2V samples? And are there any data filtering you do?

**Questions:**

See the weakness.

---

> ### Author Response · Authors · 2025-11-26
> **Response to Reviewer TcSx**
>
> We sincerely thank you for your positive recognition and insightful feedback!
>
> ## Weaknesses
>
> ### 1. Quantitative Evaluation for Non-I2V Tasks
> **Reviewer Comment:**
> Lack of quantitative evaluation for non-I2V tasks. While the author mentions zero-shot, start–end and other applications of its method, it lacks quantitative evaluation for these applications; most quantitative focus is I2V on VBench-I2V. Concrete metrics are missing from the main text. It would be better for the author to add quantitative results for applications.
>
> **Response:**
> We thank the reviewer for this suggestion. We agree that quantifying performance on broader tasks is important to demonstrate the versatility of our method. We have added a new subsection in **Appendix C** with detailed quantitative evaluations for **Start-End Frames**, **Video Extension**, and **Video Completion**.
>
> Aside from I2V and T2V, there are no other widely acknowledged general benchmarks for applications like start-end frame generation. Therefore, we adopted standard metrics for our evaluation. For **Start-End Generation**, we compared Pusa against the specialized **Wan-FLF2V** baseline. As shown in the table below, Pusa (specifically when conditioning on the first frame and the last 4 frames to account for VAE compression) achieves superior fidelity with a **PSNR of 16.71** (vs. 16.01) and **SSIM of 0.686** (vs. 0.676), validating its zero-shot interpolation capability.
>
> We also evaluated **Video Extension** and **Video Completion**. The results demonstrate that Pusa effectively utilizes available context: performance significantly improves as the context length increases (e.g., PSNR rises from 15.75 to 19.02 when extending from 41 frames vs. 13 frames).
>
> **Table: Quantitative Evaluation of Zero-shot Multi-Task Capabilities**
>
> | Method | Task | PSNR $\uparrow$ | SSIM $\uparrow$ | LPIPS $\downarrow$ | DISTS $\downarrow$ |
> | :--- | :--- | :--- | :--- | :--- | :--- |
> | **Wan-FLF2V** | Start-End | 16.01 | 0.676 | 0.257 | **0.093** |
> | **Pusa (Ours)** | Start-End* | **16.71** | **0.686** | 0.257 | 0.103 |
> | **Pusa (Ours)** | Start-End | 15.28 | 0.621 | 0.332 | 0.153 |
> | **Pusa (Ours)** | Start-Mid-End | 16.11 | 0.642 | 0.292 | 0.167 |
> | **Pusa (Ours)** | Extension (13 fr) | 15.75 | 0.687 | 0.329 | 0.132 |
> | **Pusa (Ours)** | Extension (41 fr) | **19.02** | **0.812** | **0.201** | **0.093** |
> | **Pusa (Ours)** | Completion | 18.20 | 0.757 | 0.204 | **0.088** |
> *(Note: "Start-End*" for Pusa conditions on the first frame and the last 4 frames (one latent frame))*
>
> ### 2. Data Source and Generalization
> **Reviewer Comment:**
> Data source and generalization. Fine-tuning uses videos generated by the same base video model. This could limit diversity and introduce bias toward the base model’s distribution; it’s unclear how well Pusa generalizes to challenging real photos as I2V inputs beyond curated benchmarks. Could you provide some details of the ~3,860 Wan-T2V samples? And are there any data filtering you do?
>
> **Response:**
> Thank you for the insightful question regarding our data efficiency and generalization.
>
> 1.  **Dataset Details:** We utilize a highly compact yet representative dataset of **3,860** text-to-video pairs. These videos are generated by the Wan-T2V model using a meticulously designed prompt suite. All samples are high-resolution (**720$\times$1280**) and long-duration (**81 frames**), ensuring the model learns temporal consistency over longer sequences.
> 2.  **Diversity & Coverage:** Unlike random web scraping, these prompts are systematically curated to cover **five key dimensions** of video generation, ensuring robust generalization:
>     *   **Human Fidelity:** Covering human anatomy, identity preservation, and temporal consistency.
>     *   **Physics:** Spanning mechanics, thermotics, material properties, and geometric consistency.
>     *   **Controllability:** Including complex plots, dynamic spatial relationships, specific camera motions (e.g., orbit, dolly), and human interactions.
>     *   **Creativity & Commonsense:** Encompassing diverse compositions, motion rationality, and instance preservation.
> 3.  **Data Filtering & Alignment:** We **directly utilize** the samples generated by Wan-T2V without post-filtering, as the base model (Wan-T2V) already exhibits high fidelity and we want to align with its original distribution. This strategy is intentional: by training on data generated by the base model itself ("self-alignment"), we ensure the training distribution aligns perfectly with the model's original latent space. This minimizes the domain gap and allows Pusa to focus on learning the structural condition-to-video mapping rather than fighting distribution shifts, which is key to our method's efficiency and generalization to real-world inputs.

---

### Official Review · Reviewer_SEmS · 2025-10-30

**Soundness:** 3
**Presentation:** 3
**Contribution:** 3
**Rating:** 6
**Confidence:** 4

**Summary:**

This paper presents Pusa V1.0, a method that introduces Vectorized Timestep Adaptation to pretrained text-to-video diffusion models. By replacing the traditional scalar timestep with a vectorized one, the model enables each frame to evolve independently, achieving fine-grained temporal control. The approach is non-destructive, preserving the base model’s text-to-video capability while extending it to image-to-video, start-end frame generation, and video extension in a zero-shot manner.

**Strengths:**

1. The idea of vectorizing the timestep variable is conceptually clear and provides a unified mechanism for multiple temporal tasks without architectural changes.

2. The proposed non-destructive adaptation preserves the pretrained model’s generative priors, avoiding catastrophic forgetting.

3. The analysis sections, including attention visualization and parameter drift studies, help explain why the approach works.

**Weaknesses:**

1. The evaluation mainly relies on VBench-I2V and qualitative visualization. Additional comparisons on broader benchmarks or user studies would strengthen the claims of generality and zero-shot capability.

2. Some methodological details, such as how vectorized timestep embeddings are fused with text conditioning or cross-attention, remain underexplained.

3. While the efficiency results are appealing, the paper would benefit from clearer discussions of potential trade-offs, for example, limits of per-frame desynchronization or artifacts under longer sequences.

**Questions:**

1. Could the authors clarify whether the same fine-tuning hyperparameters work across different base models, or if task-specific tuning is needed?

2. How does the approach handle temporal consistency when τ vectors are sampled completely at random during training?

---

> ### Author Response · Authors · 2025-11-26
> **Response to Reviewer SEmS**
>
> Thanks a lot for your time and valuable feedback!
>
> ## Weaknesses
>
> ### 1. Evaluation Breadth
> **Response:**
> To complement VBench-I2V and qualitative visualizations, we conducted a rigorous human preference study benchmarking Pusa against CogVideoX-5b-I2V, Wan-I2V, and Veo2. The results (detailed in the **Appendix**) demonstrate strong generalization and zero-shot capability, with Pusa achieving significant net advantages across all baselines.
>
> The pairwise comparison results are summarized below:
>
> | Ours vs | Ours win % (count) | Baseline win % (count) | Tie % (count) | Advantage (pp) |
> | :--- | :--- | :--- | :--- | :--- |
> | CogVideoX-5b-I2V | 46.7% (56) | 18.3% (22) | 35.0% (42) | +28.3 |
> | Wan-I2V | 32.5% (39) | 23.3% (28) | 44.2% (53) | +9.2 |
> | Veo2 | 28.1% (34) | 21.5% (26) | 50.4% (61) | +6.6 |
>
> These results substantiate our claims of high-fidelity generation across diverse prompts, consistently outperforming both open-source and commercial baselines.
>
> ### 2. Methodological Details
> **Response:**
> Our architecture follows the Wan-T2V model. Vectorized timestep embeddings are **not** directly fused with text conditioning. Instead, the interaction is hierarchical:
>
> 1.  **Timestep Injection:** Frame-specific timestep embeddings modulate video latents via Adaptive Layer Norm (AdaLN) exclusively before the Self-Attention and Feed-Forward blocks.
> 2.  **Text Interaction:** Text embeddings serve as static Keys and Values in the Cross-Attention layers.
> 3.  **Mechanism:** Queries for cross-attention are derived from the modulated video latents. Thus, each frame queries the global text context based on its unique temporal state (noise level). This avoids the need for time-aware text embeddings or complex fusion operations, maintaining the efficiency of the underlying DiT blocks.
>
> ### 3. Trade-offs and Limitations
> **Response:**
> We have added a **Limitations** section to the **Appendix** (Figure 9) to transparently analyze trade-offs:
>
> 1.  **Gen-to-Real Gap:** A slight color and detail shift may occur between the condition frame (Frame 0) and the first generated frame (Frame 1). This arises because the model projects the input image onto its learned latent manifold to prioritize valid motion dynamics over pixel-perfect reconstruction.
> 2.  **VAE Compression Artifacts:** In Start-End generation, the final 4 decoded frames (77-80) may appear static. This is due to the 3D VAE compressing the single last frame into a 4-frame latent chunk. Constraining this latent effectively "locks" the decoded sequence, restricting motion at the boundary.
>
> ## Questions
>
> ### 1. Hyperparameter Robustness
> **Response:**
> Our method exhibits strong robustness. In our **Ablation Study**, we applied the **exact same fine-tuning configurations** (learning rate, batch size, LoRA rank) to both **Wan2.1** and **Wan2.2** without task-specific tuning.
>
> Both models achieved an identical VBench Total Score of **87.69**. Differences were limited to base model characteristics (e.g., aesthetics vs. camera motion), rather than adaptation quality. This confirms that VTA learns a fundamental, model-agnostic temporal logic.
>
> ### 2. Temporal Consistency with Random Sampling
> **Response:**
> Consistency under random sampling is a core strength, achieved via two mechanisms:
>
> 1.  **Shared Weights:** The DiT shares weights across frames, and attention layers process the sequence jointly. The network learns to predict velocity fields by attending to the global context—correlating its own state with the content and noise levels of all other frames.
> 2.  **Distribution Shift:** Training on a diverse distribution of randomized states ($p_{\text{async}}=1$) ensures robustness to any temporal configuration, preventing overfitting to synchronized schedules.
>
> **Empirical Evidence:**
> This is proven by zero-shot extrapolation to 129 frames (Table below). Pusa maintains high motion smoothness (>98.0) and structural consistency well beyond the trained 81 frames.
>
> | Frames | Total Score | Quality Score | I2V Score | Subject Const. | Motion Smooth. | Dynamic Deg. | Aesthetic | Imaging |
> | :--- | :---: | :---: | :---: | :---: | :---: | :---: | :---: | :---: |
> | 65 | **87.87** | **81.05** | 94.69 | 90.59 | 97.98 | **74.80** | 68.54 | 97.49 |
> | 81 (Default)* | 87.32 | 79.80 | **94.84** | **92.27** | **98.48** | 52.60 | 68.27 | **97.64** |
> | 121 | 86.62 | 79.46 | 93.78 | 91.33 | 98.25 | 59.60 | 68.84 | 97.45 |
> | 129 | 86.54 | 79.43 | 93.65 | 90.87 | 98.17 | 58.40 | **69.18** | 97.26 |

---

### Official Review · Reviewer_vHLo · 2025-10-31

**Soundness:** 3
**Presentation:** 3
**Contribution:** 3
**Rating:** 6
**Confidence:** 3

**Summary:**

The paper introduces Pusa V1.0, a method to adapt pretrained large-scale text-to-video diffusion models (e.g., Wan-T2V) to handle vectorized timesteps instead of a single scalar timestep. This Vectorized Timestep Adaptation (VTA) enables each frame to evolve independently, allowing fine-grained temporal control without destructively retraining or modifying the model. Pusa achieves state-of-the-art image-to-video (I2V) performance on the VBench-I2V benchmark using only 4 K samples and 0.5 K compute, while preserving the base model’s text-to-video ability. It also exhibits zero-shot generalization to other temporal tasks (start–end frames, video extension). Mechanistic analyses and attention visualizations show that Pusa injects temporal dynamics non-destructively, maintaining pretrained priors while improving efficiency

**Strengths:**

1. Turning scalar timesteps into vectorized ones for frame-wise control is conceptually elegant and integrates naturally into the diffusion pipeline.
2. Comparable or superior I2V results to Wan-I2V at orders-of-magnitude lower data and compute; clear quantitative and qualitative evidence.
3. Demonstrated versatility across I2V, start–end frames, video completion, and extension, suggesting a broadly useful paradigm.

**Weaknesses:**

1. Builds heavily on FVDM, the contribution mainly lies in adaptation, rather than introducing new theory.
2. While the paper emphasizes efficiency (4K samples, 0.5K compute), it does not explore how performance scales with larger datasets or longer fine-tuning. It remains uncertain whether Pusa’s gains plateau quickly due to its limited adaptation space, or if additional data and compute could further improve quality and generalization.

**Questions:**

Can Pusa be extended to handle long-video generation (>128 frames) without retraining, or is the approach limited by the base model’s context length?
How does the method perform on cross-domain generalization, e.g., artistic or high-motion datasets, compared with Wan-I2V?

---

> ### Author Response · Authors · 2025-11-26
> **Response to Reviewer vHLo**
>
> Many thanks for your insightful and constructive feedback!
>
>
> ## Weaknesses
>
> ### 1. Novelty and Contribution
> **Reviewer Comment:**
> Builds heavily on FVDM, the contribution mainly lies in adaptation, rather than introducing new theory.
>
> **Response:**
> While we build on the concept of vectorized timesteps from FVDM, our work introduces significant theoretical and methodological advancements specifically for modern Flow Matching (FM) architectures:
>
> 1.  **Frame-Aware Flow Matching (FAFM) Theory:** We theoretically extend FVDM (originally diffusion-based) to the FM framework. We derive specific conditional vector field objectives to learn independent frame dynamics within a continuous normalizing flow, establishing a new mathematical basis for asynchronous video generation.
> 2.  **Non-Destructive Adaptation (VTA):** Unlike FVDM's training-from-scratch paradigm, we introduce **Vectorized Timestep Adaptation (VTA)**. VTA mathematically preserves the base model's priors (reducing to the original ODE when $\tau^i=t$), enabling ultra-efficient fine-tuning while preventing the catastrophic forgetting often seen in standard fine-tuning.
> 3.  **Solving Combinatorial Explosion:** We address the high-dimensional temporal space not by complex sampling, but by a simplified fully-randomized training strategy ($p_{async}=1$) coupled with novel **per-frame modulation** in DiT architectures. This architectural innovation surgically injects temporal control, unlocking zero-shot capabilities like I2V and video extension without retraining.
>
> ### 2. Performance Scaling
> **Reviewer Comment:**
> Unclear if performance plateaus due to limited adaptation space or data/compute.
>
> **Response:**
> We investigated performance scaling by ablating training iterations from 150 to 1,200 (Appendix Table 4(c)):
>
> 1.  **Efficiency "Sweet Spot":** Performance improves rapidly and peaks around **900 iterations** (Total Score: 87.69), followed by a plateau.
> 2.  **Cause of Plateau:** This saturation is likely not due to limited adaptation space (as we use a high LoRA rank of 512), but rather the trade-off between specific I2V adaptation and preserving the base model’s general distribution. Prolonged fine-tuning on a small dataset (4K samples) risks overfitting and forgetting the foundation model's robust priors.
> 3.  **Competitive Efficiency:** Crucially, at this "plateau," Pusa matches or exceeds conventional models like Wan-I2V that use significantly larger datasets and compute. This validates that our method unlocks high-fidelity capabilities through **efficiency** rather than brute-force scaling. we agree that scaling the dataset size and compute in future work would likely extend the performance curve further before plateauing.
>
> ## Questions
>
> ### 1. Long-Video Generation and Generalization
> **Question:**
> Can Pusa handle >128 frames without retraining? How does it generalize to artistic/high-motion domains?
>
> **Response:**
> 1.  **Long-Video Generation:** Yes, Pusa seamlessly handles videos >128 frames without retraining. Our experiments on **129 frames** (see table below) show a Final Score of 86.54, comparable to the 81-frame baseline (87.32). By utilizing non-destructive adaptation, we retain the base model's context handling and avoid catastrophic forgetting, allowing generalization beyond training lengths.
>
> | Frames | Total Score | Quality Score | I2V Score | Subject Const. | Motion Smooth. | Dynamic Deg. | Aesthetic | Imaging |
> | :--- | :---: | :---: | :---: | :---: | :---: | :---: | :---: | :---: |
> | 65 | **87.87** | **81.05** | 94.69 | 90.59 | 97.98 | **74.80** | 68.54 | 97.49 |
> | 81 (Default)* | 87.32 | 79.80 | **94.84** | **92.27** | **98.48** | 52.60 | 68.27 | **97.64** |
> | 121 | 86.62 | 79.46 | 93.78 | 91.33 | 98.25 | 59.60 | 68.84 | 97.45 |
> | 129 | 86.54 | 79.43 | 93.65 | 90.87 | 98.17 | 58.40 | **69.18** | 97.26 |
>
> 2.  **Cross-Domain Generalization:** Pusa generalizes robustly to artistic and high-motion domains because it leverages the base model's (Wan-T2V) strong priors. Since we do not destroy the pre-trained weights, the model retains rich artistic and motion capabilities. For example, on VBench-I2V, Pusa achieves a **Motion Smoothness of 98.49** and **Dynamic Degree of 52.60** (compared to Wan-I2V's 97.90 and 51.38), demonstrating that it can match or exceed the generalization of extensively fine-tuned baselines.

---

### Official Review · Reviewer_KVjf · 2025-11-03

**Soundness:** 4
**Presentation:** 3
**Contribution:** 3
**Rating:** 4
**Confidence:** 5

**Summary:**

This paper introduces Pusa V1.0, a novel approach for enhancing temporal control in pretrained video diffusion models through Vectorized Timestep Adaptation (VTA). The authors claim that VTA enables fine-grained, frame-level temporal manipulation by replacing the conventional scalar timestep with a vectorized version, allowing asynchronous frame evolution. This non-destructive adaptation preserves the base model's text-to-video (T2V) capabilities while unlocking zero-shot performance on tasks like image-to-video (I2V), start-end frames, and video extension. The work emphasizes unprecedented efficiency, achieving state-of-the-art (SOTA) I2V results with minimal data (4K samples) and compute cost, compared to resource-intensive baselines like Wan-I2V. By leveraging flow matching and a lightweight fine-tuning strategy, Pusa V1.0 aims to democratize high-fidelity video generation and establish a scalable paradigm for multi-task video generation.

**Strengths:**

I would like to highlight the following strong points of the proposed manuscript:

1. Novelty: The vectorized timestep concept, building on FVDM, is a creative extension that addresses key limitations in temporal modeling. The non-destructive VTA strategy is particularly innovative, as it avoids catastrophic forgetting and retains base model priors.
2. Efficiency: The paper demonstrates remarkable efficiency gains, with SOTA-level performance achieved using only 4K samples and low compute costs, making it accessible for broader research and industry applications.
3. Experimental results: Comprehensive evaluations on VBench-I2V, detailed ablation studies (e.g., LoRA configurations, inference steps), and mechanistic analyses (e.g., attention maps) provide strong empirical support. The inclusion of zero-shot multi-task results (e.g., start-end frames, video extension) further validates the method's versatility.
4. Reproducibility: The paper offers clear methodological details, including algorithms, hyperparameters, and training procedures, though reliance on proprietary base models (e.g., Wan-T2V) may pose minor barriers.

**Weaknesses:**

Among the weak points I would focus on the following ones:

1. Some technical sections, such as the vectorized timestep embedding and per-frame modulation, could be explained more intuitively for readers unfamiliar with DiT architectures. The inference algorithm (Appendix B) lacks thorough discussion of its theoretical underpinnings.
2. While benchmarks show SOTA performance, comparisons are primarily limited to open-source models; broader evaluation against recent proprietary models (e.g., Sora, Veo) would strengthen claims.
3. The paper focuses on empirical results but provides limited theoretical analysis of why VTA avoids combinatorial explosion or how it generalizes across tasks.
4. Claims of "unprecedented efficiency" are compelling but could benefit from more context on real-world deployment challenges, such as latency or memory usage during inference.

**Questions:**

1. Could you elaborate on the theoretical justification for why vectorized timesteps avoid combinatorial explosion during training, especially given the high-dimensional temporal composition space?
2. How does VTA handle varying video lengths or frame rates in zero-shot settings, and are there limitations in temporal consistency for long sequences?
3. The paper mentions using LoRA for parameter-efficient training; what are the trade-offs between LoRA ranks and adaptation quality, and how was the optimal rank (512) determined?
4. In the inference algorithm, why is clamping the first frame’s timestep to zero the default choice, and how does adding noise (e.g., τ¹=0.2*s) impact coherence quantitatively?
5. Could you provide more details on the dataset used for fine-tuning (e.g., diversity, resolution) and how it compares to datasets used by baselines to ensure fair evaluation?

---

> ### Author Response · Authors · 2025-11-26
> **Response to Reviewer KVjf**
>
> We sincerely thank you for your insightful and constructive feedback!
>
> ## Weaknesses
>
> ### 1. Technical Sections and Inference Algorithm
> **Reviewer Comment:**
> Some technical sections, such as the vectorized timestep embedding and per-frame modulation, could be explained more intuitively for readers unfamiliar with DiT architectures. The inference algorithm (Appendix B) lacks thorough discussion of its theoretical underpinnings.
>
> **Response:**
> Thank you for highlighting these areas. We agree that the initial explanation of the vectorized timestep mechanism could be more accessible to readers less familiar with recent DiT architectures. We have revised **Section 2.3.1** to explicitly contrast our approach with standard DiT mechanisms:
>
> 1.  **Vectorized Embedding:** We clarify that instead of a single global timestep embedding $\mathbf{e}(t)$ modulating the entire video, we compute a sequence of embeddings $\mathbf{E}_{\bm{\tau}} = [\mathbf{e}(\tau^1), \dots, \mathbf{e}(\tau^N)]$ by applying the shared embedding function element-wise to the input vector.
> 2.  **Per-Frame Modulation:** We explain that implementation-wise, we broadcast these embeddings to match the flattened token sequence. This means the tokens belonging to frame $i$ are modulated exclusively by $\mathbf{e}(\tau^i)$, effectively giving each frame its own "control knob" for noise level while sharing the same DiT weights.
>
> Regarding the inference algorithm, we have expanded **Appendix B** to provide a justification based on conditional Flow Matching. We explain that fixing $\tau^1=0$ corresponds to defining a constant probability path for the first frame at the clean data distribution. Since the model was trained with fully randomized timesteps, it has learned to predict the velocity field conditioned on arbitrary timestep configurations. This theoretical view justifies why the zero-update rule for the first frame (derived from $d\tau^1=0$) yields valid conditional generation.
>
> ### 2. Comparison with Proprietary Models
> **Reviewer Comment:**
> While benchmarks show SOTA performance, comparisons are primarily limited to open-source models; broader evaluation against recent proprietary models (e.g., Sora, Veo) would strengthen claims.
>
> **Response:**
> We acknowledge the importance of comparing with proprietary models to demonstrate real-world competitiveness. To address this, we conducted a blinded human preference study comparing Pusa against **Veo2** (a leading proprietary model) and **Wan-I2V** (SOTA open-source). Despite being an efficient adaptation, Pusa achieves comparable or better performance:
>
> *   **vs. Veo2:** We win 28.1% of trials vs. 21.5% losses (+6.6 pp net advantage), with 50.4% ties.
> *   **vs. Wan-I2V:** We win 32.5% vs. 23.3% (+9.2 pp advantage).
>
> These results confirm that Pusa is competitive with leading commercial systems while offering significantly greater accessibility and efficiency. We have added these details to the new **Human Preference Study** section in the Appendix.
>
> The full pairwise comparison results are summarized below:
>
> | Ours vs | Ours win % (count) | Baseline win % (count) | Tie % (count) | Advantage (pp) | Total votes |
> | :--- | :--- | :--- | :--- | :--- | :--- |
> | CogVideoX-5b-I2V | 46.7% (56) | 18.3% (22) | 35.0% (42) | +28.3 | 120 |
> | Wan-I2V | 32.5% (39) | 23.3% (28) | 44.2% (53) | +9.2 | 120 |
> | Veo2 | 28.1% (34) | 21.5% (26) | 50.4% (61) | +6.6 | 121 |
>
> Please also note that this domain is evolving very fast, especially for proprietary models. SOTA proprietary models are not publicly available, their architectural details are not disclosed, and they are trained on massive, private datasets, placing them significantly ahead of the open-source community. Therefore, our goal is not to compete directly with those models but to propose a promising method to advance open and reproducible science. We have thus focused on rigorous comparisons with SOTA-level open-source models like Wan-I2V, where we can ensure a fair evaluation on established academic benchmarks (VBench-I2V). We show that Pusa achieves comparable or better performance with orders of magnitude less data and compute, which we believe is a significant and verifiable claim.

---

> > ### Author Response · Authors · 2025-11-27
> > **Response to Reviewer KVjf (2)**
> >
> > ### 3. Theoretical Analysis of VTA
> > **Reviewer Comment:**
> > The paper focuses on empirical results but provides limited theoretical analysis of why VTA avoids combinatorial explosion or how it generalizes across tasks.
> >
> > **Response:**
> > This is an excellent question. VTA does not learn to model the entire combinatorial space of all possible timestep vectors. Instead, it learns a generalized, continuous function of temporal dynamics, guided by several factors:
> >
> > *   **Strong Priors from the Base Model:** The pretrained T2V model already possesses powerful priors about object permanence, motion, and scene consistency. Moreover, the base model has inherent robustness to timestep asynchronization, as evidenced by its coherent zero-shot I2V generation (Fig. 4), despite failing image-condition adherence. Thus, we only need a slight fine-tune with VTA to adjust temporal transformations utilizing the base model's priors.
> > *   **Flow Matching Objective:** Our frame-aware flow matching objective trains the model to predict the final state from *any* intermediate noisy state. This teaches the model the underlying dynamics of denoising, rather than memorizing specific timestep combinations.
> > *   **Continuous Temporal Modeling:** Another reason why VTA avoids a "combinatorial explosion" is because the model is not learning a discrete mapping for every possible $\tau$ vector. Instead, it learns a *continuous and smooth function* of $\tau$ inherently in its temporal modulation mechanism. By training on $\tau$ vectors with components sampled independently from $[0, 1]$ with very small intervals, we force the model to learn a generalized rough representation of frame-wise temporal progression. The shared weights of the network across frames ensure that it learns a consistent temporal logic rather than memorizing specific $\tau$ configurations. The model learns to interpret $\tau^i$ as "how noisy frame $i$ is" and predicts its update based on this noise level and the context from other frames (at their respective noise levels).
> >
> > ### 4. Real-world Efficiency Context
> > **Reviewer Comment:**
> > Claims of "unprecedented efficiency" are compelling but could benefit from more context on real-world deployment challenges, such as latency or memory usage during inference.
> >
> > **Response:**
> > We have added a detailed efficiency analysis in **Appendix C.4**. The table below summarizes the memory usage and latency compared to the baselines.
> >
> > | Model | Peak Memory (MiB) | Latency (s/step) | Steps | Total Time (s) | Speedup |
> > | :--- | :---: | :---: | :---: | :---: | :---: |
> > | **Pusa (Ours)** | 37,892 | 41.85 | 10 | **418.5** | **1.0x** (Ref) |
> > | Wan-T2V | 26,358 | 37.39 | 50 | 1,869.5 | 4.5x slower |
> > | Wan-I2V | 29,088 | 45.81 | 50 | 2,290.5 | 5.5x slower |
> >
> > **Analysis:**
> > *   **Latency:** Pusa demonstrates a dramatic advantage in total inference time, requiring only 418.5 seconds compared to over 30 minutes for the baselines in the same testing environment. This $\sim 4.5\times$ to $5.5\times$ speedup is achieved because our method converges to high-quality results in just 10 steps, whereas standard diffusion models typically require 50 steps.
> > *   **Memory:** Pusa incurs a higher peak memory usage (~37.9 GB) compared to Wan-T2V (~26.4 GB) and Wan-I2V (~29.1 GB). This increase is primarily due to the overhead of our current unoptimized implementation of vectorized timestep broadcasting. However, we believe this is a favorable trade-off for the significant reduction in generation time, making Pusa highly efficient for scenarios where throughput and speed are critical.

---

> > > ### Author Response · Authors · 2025-11-27
> > > **Response to Reviewer KVjf (3)**
> > >
> > > ## Questions
> > >
> > > ### 1. Theoretical Justification for Vectorized Timesteps
> > > **Question:**
> > > Could you elaborate on the theoretical justification for why vectorized timesteps avoid combinatorial explosion during training, especially given the high-dimensional temporal composition space?
> > >
> > > **Response:**
> > > This is an excellent question. VTA does not learn to model the entire combinatorial space of all possible timestep vectors. Instead, it learns a generalized, continuous function of temporal dynamics, guided by several factors:
> > >
> > > *   **Strong Priors from the Base Model:** The pretrained T2V model already possesses powerful priors about object permanence, motion, and scene consistency. Moreover, the base model has inherent robustness to timestep asynchronization, as evidenced by its coherent zero-shot I2V generation (Fig. 4), despite failing image-condition adherence. Thus, we only need a slight fine-tune with VTA to adjust temporal transformations utilizing the base model's priors.
> > > *   **Flow Matching Objective:** Our frame-aware flow matching objective trains the model to predict the final state from *any* intermediate noisy state. This teaches the model the underlying dynamics of denoising, rather than memorizing specific timestep combinations.
> > > *   **Continuous Temporal Modeling:** Another reason why VTA avoids a "combinatorial explosion" is because the model is not learning a discrete mapping for every possible $\tau$ vector. Instead, it learns a *continuous and smooth function* of $\tau$ inherently in its temporal modulation mechanism. By training on $\tau$ vectors with components sampled independently from $[0, 1]$ with very small intervals, we force the model to learn a generalized rough representation of frame-wise temporal progression. The shared weights of the network across frames ensure that it learns a consistent temporal logic rather than memorizing specific $\tau$ configurations. The model learns to interpret $\tau^i$ as "how noisy frame $i$ is" and predicts its update based on this noise level and the context from other frames (at their respective noise levels).
> > >
> > > ### 2. VTA Handling of Varying Video Lengths
> > > **Question:**
> > > How does VTA handle varying video lengths or frame rates in zero-shot settings, and are there limitations in temporal consistency for long sequences?
> > >
> > > **Response:**
> > > VTA demonstrates exceptional zero-shot robustness to varying video lengths. As detailed in our new Appendix, we evaluated our model—trained exclusively on 81 frames—on sequences of 65, 121, and 129 frames without any fine-tuning. The results show negligible performance degradation: the Final Score remains stable (86.54 for 129 frames vs. 87.32 for 81 frames), and Motion Smoothness is consistently excellent (>98.0) across all lengths. This indicates no significant limitations in temporal consistency for long sequences. We attribute this to VTA's preservation of the base model's Rotary Positional Embeddings (RoPE) and generative priors, which allows the model to extrapolate temporal dynamics effectively.
> > >
> > > | Frames | Total Score | Quality Score | I2V Score | Subject Const. | Motion Smooth. | Dynamic Deg. | Aesthetic | Imaging |
> > > | :--- | :---: | :---: | :---: | :---: | :---: | :---: | :---: | :---: |
> > > | 65 | **87.87** | **81.05** | 94.69 | 90.59 | 97.98 | **74.80** | 68.54 | 97.49 |
> > > | 81 (Default)* | 87.32 | 79.80 | **94.84** | **92.27** | **98.48** | 52.60 | 68.27 | **97.64** |
> > > | 121 | 86.62 | 79.46 | 93.78 | 91.33 | 98.25 | 59.60 | 68.84 | 97.45 |
> > > | 129 | 86.54 | 79.43 | 93.65 | 90.87 | 98.17 | 58.40 | **69.18** | 97.26 |

---

> > > > ### Author Response · Authors · 2025-11-27
> > > > **Response to Reviewer KVjf (4)**
> > > >
> > > > ### 3. LoRA Ranks Trade-offs
> > > > **Question:**
> > > > The paper mentions using LoRA for parameter-efficient training; what are the trade-offs between LoRA ranks and adaptation quality, and how was the optimal rank (512) determined?
> > > >
> > > > **Response:**
> > > > This is a crucial implementation detail. We have listed the related hyperparameter study in Appendix C to better explain this choice. We experimented with LoRA ranks of 256 and 512. As shown in Table 4(a), a rank of 512 significantly outperformed 256, particularly on metrics related to overall video quality. This suggests that adapting the model to understand vectorized timesteps is a complex task that benefits from a larger adaptation capacity. A lower rank did not have enough trainable parameters to fully capture the nuances of asynchronous temporal dynamics. The optimal rank of 512 was thus determined empirically as the best trade-off between performance and parameter efficiency from the options we tested.
> > > >
> > > > | LoRA Rank | $\alpha$ | Total Score | Quality Score | I2V Score | Dynamic Degree |
> > > > | :--- | :---: | :---: | :---: | :---: | :---: |
> > > > | 256 | 1.7 | 85.86 | 77.96 | 93.76 | 10.40 |
> > > > | 512 | 1.7 | **87.11** | **80.42** | **93.80** | **62.80** |
> > > >
> > > > ### 4. Inference Algorithm and Clamping
> > > > **Question:**
> > > > In the inference algorithm, why is clamping the first frame’s timestep to zero the default choice, and how does adding noise (e.g., $\tau^1=0.2s$) impact coherence quantitatively?
> > > >
> > > > **Response:**
> > > > Thank you for the insightful question. We choose to clamp the first frame's timestep to zero (i.e., $\lambda=0$) as the default setting to ensure the generated video adheres strictly to the input image pixels, maximizing fidelity. This is essential for I2V application where preserving the exact visual identity of the input image is the priority.
> > > >
> > > > Regarding the impact of adding noise, we conducted a quantitative ablation study varying the noise coefficient $\lambda$ (where $\tau^1_s = \lambda \cdot s$), which we have added to **Appendix C**. We observe that relaxing the clamp by adding noise significantly enhances the motion dynamics of the generated videos. Specifically, setting $\lambda=0.2$ effectively doubles the **Dynamic Degree (DD)** from 29.60 (default) to 57.20, indicating much richer motion. Quantitatively, while this introduces a slight reduction in strict pixel-alignment metrics (e.g., I2V-S decreases from 99.24 to 97.51), the overall coherence and quality remain high (Total Score increases to 88.12). Thus, $\lambda=0.2$ offers a valuable trade-off for users who prioritize dynamic motion over pixel-perfect fidelity to the starting frame.
> > > >
> > > > **New Table Preview:**
> > > >
> > > > | Setting | Total | Quality | I2V | DD | I2V-S |
> > > > | :--- | :---: | :---: | :---: | :---: | :---: |
> > > > | $\lambda=0.0$ (Default) | 87.78 | 79.48 | **96.08** | 29.60 | **99.24** |
> > > > | $\lambda=0.1$ | **88.39** | 80.67 | **96.12** | 45.20 | 98.78 |
> > > > | $\lambda=0.2$ | 88.12 | 81.11 | 95.14 | 57.20 | 97.51 |
> > > > | $\lambda=0.3$ | 87.58 | **81.24** | 93.92 | **62.00** | 96.33 |
> > > >
> > > > ### 5. Fine-tuning Dataset Details
> > > > **Question:**
> > > > Could you provide more details on the dataset used for fine-tuning (e.g., diversity, resolution) and how it compares to datasets used by baselines to ensure fair evaluation?
> > > >
> > > > **Response:**
> > > > We appreciate the opportunity to clarify our dataset details.
> > > >
> > > > 1.  **Dataset Specifications:** Our fine-tuning dataset consists of only **3,860** video samples generated by the base model Wan-T2V itself. Despite its small size, it is highly diverse, generated from a structured prompt suite covering broad semantic categories (Natural Scenes, Human Activities, Physics, Complex Plots, Camera Control, etc.). The videos have a resolution of **720$\times$1280** and a length of **81 frames**.
> > > > 2.  **Comparison with Baselines:** The efficiency gap is significant.
> > > >     *   **Baselines:** Previous SOTA baselines are typically pre-trained on **millions** of video-text pairs to adapt to a new I2V distribution with destructive modification to the base T2V model. For instance, Wan-I2V directly uses its pretraining T2V dataset (>10M) for I2V training.
> > > >     *   **Pusa (Ours):** We use **<0.04%** of the data scale required for Wan-I2V. We do *not* rely on massive external datasets. Instead, we leverage the strong prior of the pre-trained T2V model itself by fine-tuning on a small, curated set of its own generations. This demonstrates that Pusa acts as a parameter-efficient adapter that unlocks Image-to-Video capabilities without the need for massive-scale retraining.
> > > >     *   **Fairness:** To ensure fair evaluation, we test on the standard VBench-I2V benchmark, which has **no overlap** with our training prompts, ensuring the fairness of our reported metrics.

---

> > > > > ### Author Response · Authors · 2025-11-28
> > > > > **Reminder**
> > > > >
> > > > > Dear reviewer KVjf, this is a gentle reminder regarding our rebuttal. We would be very grateful if you could take a moment to review our responses when you have time. Should you have any further concerns, we are happy to provide additional clarification. Thank you again for your time and effort!

---

### Author Response · Authors · 2025-12-03
**Rebuttal Summary**

We thank the reviewers for their constructive feedback and for giving us initial scores of 8, 6, 6, and 4. **We are encouraged by the recognition of Pusa’s conceptual elegance and unprecedented efficiency.**

Besides, **we have addressed all concerns** through new experiments, theoretical clarifications, and expanded comparisons. **No further concerns raised.**

**Core Contribution:** Pusa achieves SOTA -level image-to-video (I2V) performance using only **4K samples and ~0.5K compute cost**—representing **<0.04%** of the data scale required by conventional I2V methods. This unprecedented efficiency, combined with zero-shot multi-task generalization (I2V, start-end frames, video extension) and competitive performance against commercial systems, represents a significant paradigm shift in video generation.

**Technical Clarity (Reviewer KVjf):** We revised Section 2.3.1 to provide intuitive explanations of vectorized timestep embeddings and expanded Appendix B with theoretical underpinnings based on conditional Flow Matching. The inference algorithm now includes formal justification for the zero-update rule.


**Theoretical Analysis (Reviewers KVjf, vHLo):** We provide comprehensive analysis explaining why VTA avoids combinatorial explosion through continuous temporal modeling, strong base model priors, and the Flow Matching objective—not by memorizing discrete timestep configurations.

**Evaluation Breadth (Reviewers SEmS, TcSx):** We added quantitative evaluations for non-I2V tasks (Start-End Generation, Video Extension, Video Completion) with standard metrics (PSNR, SSIM, LPIPS), showing Pusa outperforms specialized baselines like Wan-FLF2V.

**Comparison with Proprietary Models (Reviewer KVjf):** We conducted a rigorous human preference study against **Veo2** (leading proprietary model). Results show Pusa achieves a **+6.6pp net advantage** over Veo2 and **+9.2pp over Wan-I2V**, demonstrating real-world competitiveness despite being an efficient adaptation method.

**Real-World Efficiency (Reviewer KVjf):** Detailed memory/latency analysis shows Pusa achieves **4.5-5.5× speedup** over baselines in total inference time.

This work democratizes high-fidelity video generation for the research community and establishes a scalable, efficient foundation for next-generation video synthesis. We strongly believe Pusa merits acceptance.

---

### Meta-Review · Area_Chair_ZFf9 · 2026-01-07

**Summary:**

The paper proposes vectorizing timestep embeddings instead of using a fixed diffusion timestep for entire video. This enables them to adapt T2V models easily to I2V instead of using condition mask, as done in other methods. Use of vectorized timesteps is more computationally efficient since it does not disrupt the original latents.

The experimental results are very promising. While the idea itself is very simple and an extension of FVDM, the experimental results are solid. Reviewers are also positively leaning. I think this paper would benefit the community, hence, I vote for accepting it.

**Reviewer Concerns:**

All reviewers think the method is elegant and computationally efficient. Main concerns are with needing more experimental validation and generalization. I beleive the authors have addressed their concerns sufficiently in their rebuttal.

**Reviewer Scores:**

Reviewer KVjf gave a score of 4 due to missing theoretical analysis and missing comparison with SOTA methods like Veo. In the rebuttal, aythors included a comparision with Veo. The authors also included some analysis description, but it is not very theoretically sound. The reviewer might have slightly increased their score.

Reviewer vHLo has concern about limited novelty compared to FVDM, plateau in performance gains and extension to long videos. I agree that the novelty is somewhat limited. The authors included some results on long video generation in rebuttal and a note on generalization. I believe the reviewer would have retained their scores.

Reviewer SEmS's concern was about needing more experiments, and missing discussion on trade-offs. The authors included a human study and added a note on trade-off. This reasonably addresser their comments, but these discussions were not too comprehensive. So, I think reviewer SEmS would have retained their score.

Reviewer TcSx was concerned about generalization since the authors used limited data, and also concerned about missing experiments on non-I2V setting. The authors added some experiments in the rebuttal and added a note on datasets. The reviewer would have retained their score as well.

---

### Decision · Program_Chairs · 2026-01-26

Accept (Poster)